# AI enabled sign language recognition and VR space bidirectional communication using triboelectric smart glove

Feng Wen[1,2,3,5], Zixuan Zhang[1,2,3,5], Tianyiyi He[1,2,3] & Chengkuo Lee ⬤ [1,2,3,4 ✉]

Sign language recognition, especially the sentence recognition, is of great significance for lowering the communication barrier between the hearing/speech impaired and the non-signers. The general glove solutions, which are employed to detect motions of our dexterous hands, only achieve recognizing discrete single gestures (i.e., numbers, letters, or words) instead of sentences, far from satisfying the meet of the signers' daily communication. Here, we propose an artificial intelligence enabled sign language recognition and communication system comprising sensing gloves, deep learning block, and virtual reality interface. Non-segmentation and segmentation assisted deep learning model achieves the recognition of 50 words and 20 sentences. Significantly, the segmentation approach splits entire sentence signals into word units. Then the deep learning model recognizes all word elements and reversely reconstructs and recognizes sentences. Furthermore, new/never-seen sentences created by new-order word elements recombination can be recognized with an average correct rate of 86.67%. Finally, the sign language recognition results are projected into virtual space and translated into text and audio, allowing the remote and bidirectional communication between signers and non-signers.

[1] Department of Electrical & Computer Engineering, National University of Singapore, Singapore, Singapore. [2] National University of Singapore Suzhou Research Institute (NUSRI), Suzhou, China. [3] Center for Intelligent Sensors and MEMS (CISM), National University of Singapore, Singapore, Singapore. [4] NUS Graduate School—Integrative Sciences and Engineering Program (ISEP), National University of Singapore, Singapore, Singapore. [5]These authors contributed equally: Feng Wen, Zixuan Zhang. ✉email: elelc@nus.edu.sg

Benefiting from a number of attractive features, such as light weight, good compliance, and desirable comfortability, wearable sensors, hold great promise in various applications spanning from environmental monitoring, personalized healthcare to soft robotics, and human–machine interaction[1–9]. Typical wearable sensors rely on resistive and capacitive mechanisms for human status tracking and surrounding sensing[10–18]. Generally, they need an external power supply to generate signal excitation, inhibiting their further widespread deployment. Owing to simple fabrication, wide material choice, and expeditious dynamic response, the triboelectric nanogenerator (TENG)-based wearable sensors[19–23], which are recognized as power-compatible and self-sustainable alternatives, are increasingly employed for healthcare monitoring, human motion tracking, human–machine interface (HMI), etc.[24–30] since its invention in 2012[31]. Among them, wearable HMIs are in burgeoning demand as a promising advanced solution for achieving human–machine even human–human (e.g., signers and nonsigners) interaction by virtue of efficient human state tracking. In addition to the general HMIs using resistive and capacitive techniques[32–37], self-powered triboelectric HMIs are also extensively investigated with different prototypes, such as touchpad, wrist band, sock, and glove[38–46]. Particularly, as a promising paradigm for HMI, the glove can seamlessly detect the multiple degrees-of-freedom motions of our dexterous hands, holding great promise for the advanced demands beyond simple control.

Combining the advantages of the glove platform (e.g., conformability, intuitiveness, and low cost) and the TENG technique (e.g., dynamic sensing effect and self-powered capability), many efforts have been devoted to the development of triboelectric glove HMI in recent years. For instance, TENG gloves were extensively demonstrated monitoring finger motions by magnitude analysis or pulse counting[47–51]. However, their data analytics are mostly based on manual simple feature extraction (e.g., amplitude, frequency, peak number), leading to a limited variety of recognizable hand motions/gestures with substantial feature loss. The sophisticated hand gesture discrimination remains challenging. Artificial intelligence (AI) has recently unlocked intelligent data analytics in interdisciplinary domains by exploiting comprehensive sensory information extraction and autonomous learning. The integration of AI with the TENG glove may shed light on more diversified and complex gesture monitoring that is rarely achievable using the conventional manual feature extraction[52]. As a proof of concept, our previous work demonstrated an AI-enabled TENG glove capable of recognizing similar and complex 11 gestures for advanced virtual reality/augmented reality (VR/AR) controls[53]. It reveals the feasibility of AI-enabled sophisticated and comprehensive hand gesture identification via minimalist TENG sensor configuration.

Being closely associated with sophisticated hand gestures and as an essential part of biomedical care, sign language interpretation is of substantial significance in bridging the gap between the hearing and speech impaired and general public since sign language is not as prevalent as the speaking language and difficult for nonsigners to understand without prior learning. Generally, visual images/videos, surface electromyography (sEMG) electrodes, and inertial sensors are conventional means to reconstruct hand gesture information towards sign language recognition applications[54–57]. They are either limited by light conditions and privacy concerns, electromagnetic noise, and crosstalk with other biopotentials, or a huge amount of data, respectively[58]. Complementary to traditional solutions, a low cost and straightforward wearable TENG glove could be a potential assistive platform for sign language perception, which is immune to the issues of traditional approaches. With the support of comprehensive feature extraction of AI, the glove renders intelligent sign language recognition that involves diversified and complicated gestures.

For instance, by leveraging AI at the cutting edge, Z. Zhou et al. initiated[59] a sign-to-speech translation system comprising a self-powered TENG glove and wireless transmission block. The system achieved a highly accurate recognition of 11 signs including numbers, letters, and a word and unidirectionally displayed recognition results on a mobile phone interface via Bluetooth, which paves the way to the practical sign translation and expectedly benefits the speech/hearing disordered. On the other hand, several simple signs in sign language can be identified without AI integration. A human-skin-inspired interlocked triboelectric sensor mounted on a glove has been reported to recognize four discrete words: 'I', 'Happy', 'Meet', and 'You'[60]. In another case, there is a developed TENG glove with only achieving five letters recognition of sign language[61]. Thus, without the assist of AI-based analytics, most current sign language translation solutions are limited to the recognition of only several discrete and simple words, numbers, or letters. Although Z. Zhou et al. demonstrated[59] the work of incorporating AI technology to realize the classification of a dozen sign gestures in a high accurate manner, it still lacks an effective and practical approach for real-time sentence recognition of sign language, which is more significant for the practical communication of signers and nonsigners. Besides, the interfaces (e.g., mobile phone or PC) that sign language recognition results are projected or displayed usually do not allow the signer's interaction with nonsigners. The VR interface has been a recent hot topic owing to enhanced interaction and immersive experience[62, 63]. It creates the potential of human-to-human (e.g., the speech/hearing disordered and healthy people) interaction, which improves the practicality of sign language recognition system.

Here, we show a sign language recognition and communication system comprising triboelectric sensor integrated gloves, AI block, and the VR interaction interface. The system successfully realizes the recognition of 50 words and 20 sentences. The recognition results are projected into virtual space in the forms of comprehensible voice and text to facilitate barrier-free communication between signers and nonsigners (Fig. 1a). First, to understanding the raw data and indicate the necessity of AI integration, the data analysis of sign sensing signals is carried out. Two frames, nonsegmentation and segmentation AI skeletons, enable the recognition of diversified words and sentences. The nonsegmentation AI framework achieves high accuracy for 50 words (91.3%) and 20 sentences (95%) by independently identifying word and sentence signals. To overcome the issue of unable to recognizing new sentences, the developed segmentation AI frame splits the entire sentence signals into word units and then recognizes all the signal fragments. Inversely, the whole sentence's information can be reconstructed and recognized with an accuracy of 85.58% upon established correlation between word units and sentences. Particularly, the segmentation approach assisted AI brings the capability of recognizing new/never-seen sentences (average correct rate: 86.67%) that are not included in the database and are created by word element recombination in new orders. It provides a promising and universal solution to recognize new/never-seen sentences and to readily expand the sentence database for practical communication of the speech/hearing disordered. Finally, the server-client terminals in VR space allow the displaying of sign recognition results and the direct typing of nonsigners. The VR interface for two-way remote communication, linked with the AI front end for sign language recognition, shows a potential prototype of a future intelligent sign language recognition and communication system.

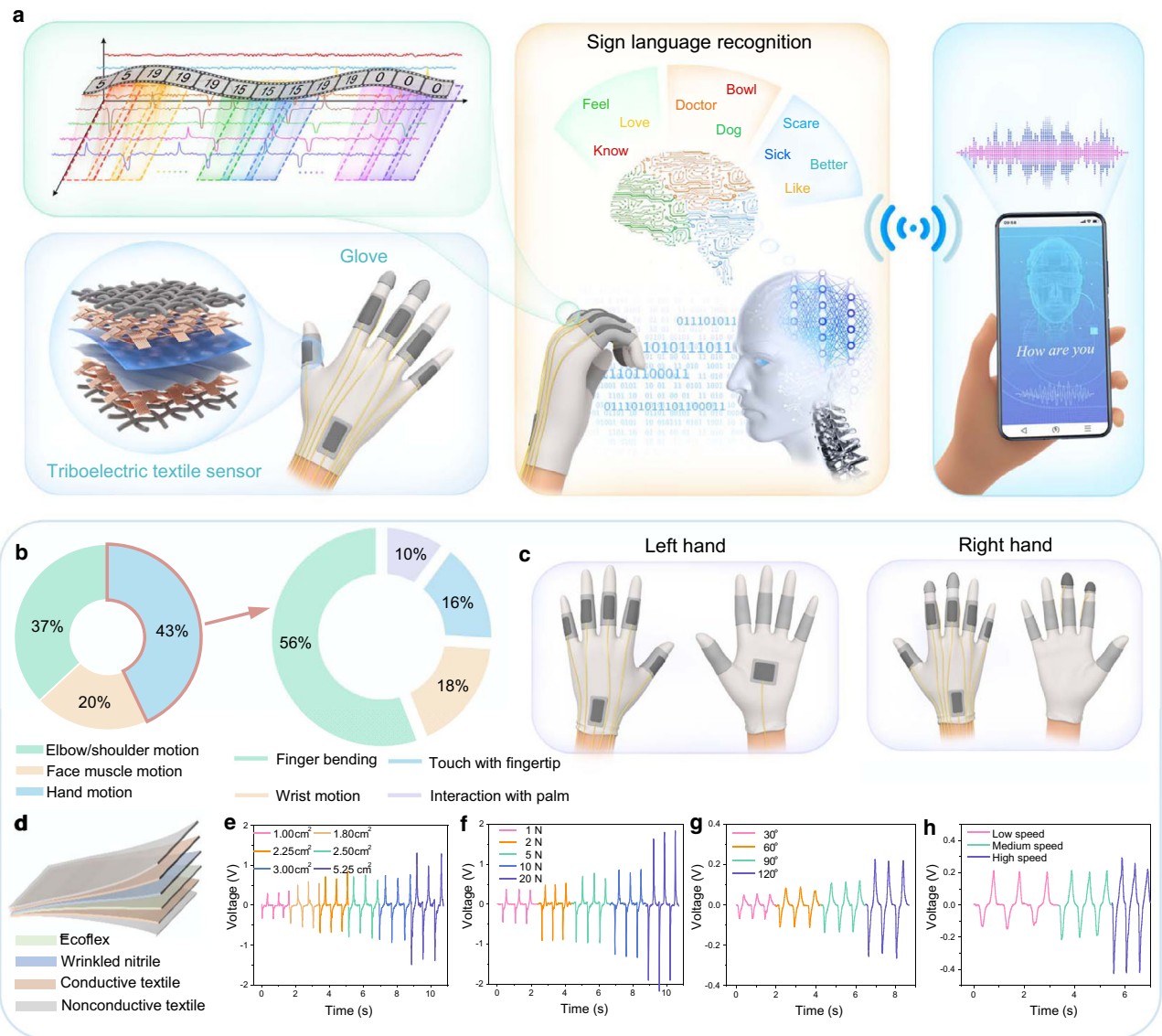

**Fig. 1 The glove configuration and sensor characterization. a** Schematics of the sign language recognition and communication system. **b** Proportion of different motions that are commonly used in sign language helps determine sensor position on gloves. **c** Sensor position on gloves based on hand motion analysis in **b**. Detailed area information of the sensor on each position can be found in Supplementary Fig. 1. **d** Materials of the triboelectric sensor. **e–h** Voltage output dependence on key parameters, including sensor area, force, bending degree, and bending speed. The hand, head, and phone images are created by the authors via Blender.

## Results

**Glove configuration and sensor characterization**. To measure as many gestures as possible with a small number of sensors, the sensor position needs to be optimized. By referring to the frequently used sign language in the American Sign Language guide book[64], we conduct the analysis for involved motions in daily sign expressions of the speech/hearing impaired. As depicted in Fig. 1b, the findings show that sign language includes three major motion dynamics, including elbow/shoulder motions, face muscle activities, and hand movements. The hand motion accounts for a proportion of 43%, remaining dominant in the three major motions, indicating the inevitability of hand motion perception in the application of sign language recognition. Further category differentiation in hand motions is necessary to confirm the sensor position, as shown in the enlarged pie chart in Fig. 1b. In detail, hand motions fall into four categories, including finger bending (56%), wrist motion (18%), touch with fingertips (16%), and interaction with palm (10%). The relevant movements need

sensors in different positions to generate the critical correspondence (Fig. 1c). Accordingly, the triboelectric sensor is mounted on each finger for finger bending measurement, while two sensors are put on wrists for wrist motion perception. Reviewing the daily used sign language shows the index and middle fingertips of the right hand are in frequent use. Meanwhile, signers often use their left hand to interact with their right hand and other parts of their body. Thus, two sensors are placed on the fingertips of right index and middle of the glove, and one sensor is incorporated on the palm of the left hand. The more detailed discussion about the sensor arrangement on the glove can be found in Supplementary Note 1.

Finally, gloves are configured with 15 triboelectric sensors in total. The area of sensors on different hand positions is customized since the coverage area is different from each hand position. The detailed area information of corresponding sensors is included in the Supplementary Fig. 1. After position optimization, the individual triboelectric sensor's basic

characterization is investigated concerning key parameters that largely influence the triboelectric voltage output performance, such as sensor area, force, bending degree, and bending speed. As presented in Fig. 1d, the triboelectrification layers are composed of ecoflex and wrinkled nitrile. With the conductive textile as electrodes, a flexible and thin triboelectric sensor is fabricated. Based on Fig. 1e, f, the voltage output increases when sensor area, force, bending degree, and bending speed increase.

**Data analysis of signals of 50 words and 20 sentences**. The tentative data analysis is beneficial for the preliminary understanding of the raw data. First, we select 50 gestures that are daily used in the signer's life as demonstration. Figure 2a shows images of representative 19 gestures. The corresponding triboelectric signals of these 19 gestures are given in Fig. 2b. The photos and signals of the rest 31 signs can be found in Supplementary Fig. 2. The signal similarity and correlation analysis of the total 50 gestures are carried out based on the original data. There are 100 data samples for each gesture in the dataset, where 'Get' is taken as the example as shown in the enlarged view in the below middle of Fig. 2b. For correlation analysis, the below left of Fig. 2b shows a new input of 'Get' compared with its own database to obtain the mean correlation coefficient $0.53 > 0.3$ (threshold value for high similarity) by extracting the average value of 100 data samples and correlating it with the new input, which means the similarity is high between different 'Get' data samples. Also, the signal of 'Must' is compared with the 'Get' database. The mean correlation coefficient of 0.37 is larger than 0.3, indicating the high signal similarity between 'Get' and 'Must' as well (the bottom right of Fig. 2b).

Figure 2c provides a matrix to summarize each gesture's correlation coefficient with the other 49 gestures. Switching such matrix information to the distribution curve of correlation coefficient depicted in Fig. 2d, it can be observed that there are signals of several gestures fallen into the strong correlation area, which means a high similarity among these gesture signals. These gestures are in the high possibility for the wrong classification. Second, 20 daily sentences are sorted out for subsequent study. Likewise, the similarity analysis among sentence signals is also conducted, as shown in Fig. 2e and Fig. 2f, which are based on the analysis of original sentence signals provided in Fig. 2g. The high similarity has been also noticed among sentences as some sentence signals have a high correlation coefficient with others. In general, the data analysis proclaims the high similarities of word-to-word and sentence-to-sentence signals. Thus, for the sign language interpretation in such a large database, a more sophisticated analysis method is in compelling demand. As a powerful technology, deep learning from AI provides a great feasibility to achieve the advanced data analytics, whose momentum exists in delicate analysis and accurate recognition by comprehensive feature extraction.

**Word and sentence recognition upon nonsegmentation method**. For sequence modeling of signals, hidden markov model (HMM), recurrent neural network (RNN), and more recently developed long short-term memory (LSTM) have been widely utilized owing to enabling memorizing the output of the last moment for circulated self-updating and adapting. They are usually applied to the construction of large-scale complex deep learning networks depending on huge data samples. Besides, the convolutional neural networks (CNN) are designed to process data in the form of multiple arrays. In particular, 1D CNN is a simple and feasible solution for recognizing the time-series signals of human motions from triboelectric sensors. Because the positions of features in the segment are not highly correlated. To

optimize the CNN model toward more effective recognition behaviors, adjustment on kernel size, the number of filters, and convolutional layers is implemented. As the results in Fig. 3a-c show, the 1D CNN model exhibits the optimal accuracy performance with 5 kernels, 64 filters, and 4 convolutional layers. The detailed CNN structure parameters are presented in Supplementary Table 1. Accordingly, the schematic diagram of the CNN structure is shown in Fig. 3d, in which the 15-channel gesture signals are served as inputs without segmentation. Then the pattern recognition is achieved through the optimized 4-layer CNN architecture. In other words, the sentence signals are not pieced into word elements and not linked with the basic word units within a nonsegmented deep learning frame. The signals of words and sentences are isolated from each other and independently recognized.

To better understand the clustering performance of the proposed CNN structure, the results of last fully connected layer is extracted in which each data sample with 3000 points finally has been stretched to 3584 after convolution and pooling to compare with the raw data. However, it is difficult to compare such high-dimensional data. The principal component analysis (PCA) is widely utilized to perform the dimension reduction and maintain the information as much as possible at the same time. By virtue of eigenvalues and eigenvectors of the analog signal matrix of gestures, which can point along the major variation directions of data, the dimension of data could be reduced with adjustable principal component matrix which comprises the nonunique eigenvectors of the data matrix. Eventually, the data of input layer and the last fully connected layer is reduced to 30 dimensionalities after PCA.

By following data visualization, Fig. 3e demonstrates the feature clustering result of 50 words for the input layer, which is achieved by the t-distributed Stochastic Neighbor Embedding (t-SNE) in the dimension of principal component 1 (PC1) and principal component 2 (PC2). It shows the performance of feature clustering is poor with category overlaps before going through the CNN network. After undergoing the feature extraction and classification of CNN, the visualization result realized by t-SNE in Fig. 3f indicates a desirable classification performance of the developed CNN model with clear boundaries among these 50 classes with less overlap. It proves the effectiveness of proposed CNN model for feature classification. After only 50 training epochs, the accuracy reaches almost 90%, as shown in Supplementary Fig. 3. As a consequence, upon 80 training samples (80%) and 20 test samples (20%) of each word in the dataset, a high recognition accuracy of these 50 words is achieved with the value of 91.3%, as shown in the confusion matrix in Fig. 3g. Supplementary Movie 1 displays the successful word recognition in Python with six representative gestures. Regarding the sentence recognition, the time-sequence signals of sentences comprise signals from several single gestures. Throwing the intact signal of each sentence with one known label into the AI processing pool, the supervised learning could clearly differentiate these 17 sentences even the sentence signal is much longer than that of words. A satisfying cluster result is also observed by comparing results in Fig. 3h and Fig. 3i. The long-length signal of sentence provides more distinguishable features than the single-word signal. Therefore, a high sentence recognition accuracy of 95% is obtained, as indicated in Fig. 3j. Correspondingly, Supplementary Movie 2 demonstrates the successful sentence recognition in which three sentences are taken as examples.

Although with high accuracies in discriminating words and sentences, the CNN model upon nonsegmentation approach only enables classifying existed sentence signals in the dataset. It cannot distinguish new/never-seen sentences even though the

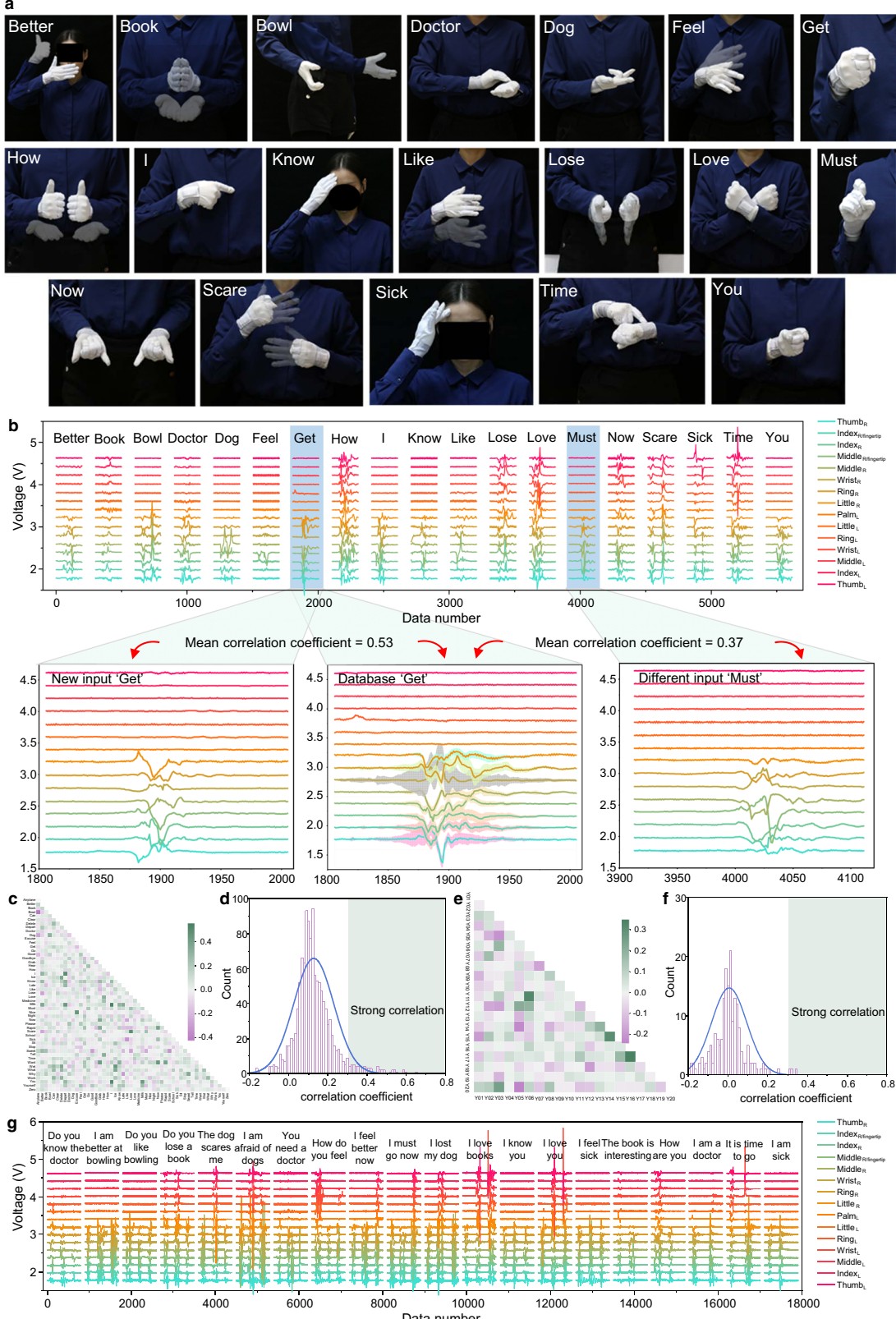

**Fig. 2 The data analysis for signals of 50 words and 20 sentences. a** Part of representatives among 50 words or gestures (showing 19 gestures here), in which the opaque and translucent gesture images show the starting and final state of the gesture, respectively. The rest of 31 gesture photo and their corresponding triboelectric signals can be found in Supplementary Fig. 2. **b** Triboelectric voltage output of 19 words (top), and the similarity and correlation analysis based on the word signals (bottom). The high correlation coefficient of 'Get' and 'Must' shows a high similarity between these two gesture signals, indicating a high possibility for wrong classification. **c** Correlation coefficient matrix of signals of 50 words. **d** Correlation coefficient distribution curve of 50 words. **e** Correlation coefficient matrix of 20 sign language sentences. **f** Correlation coefficient distribution curve of 20 sentences. **g** Voltage output of 20 sentences. Photo credit: Feng Wen, National University of Singapore. Source data[65] of Figs. 2c, e are provided in Harvard Dataverse.

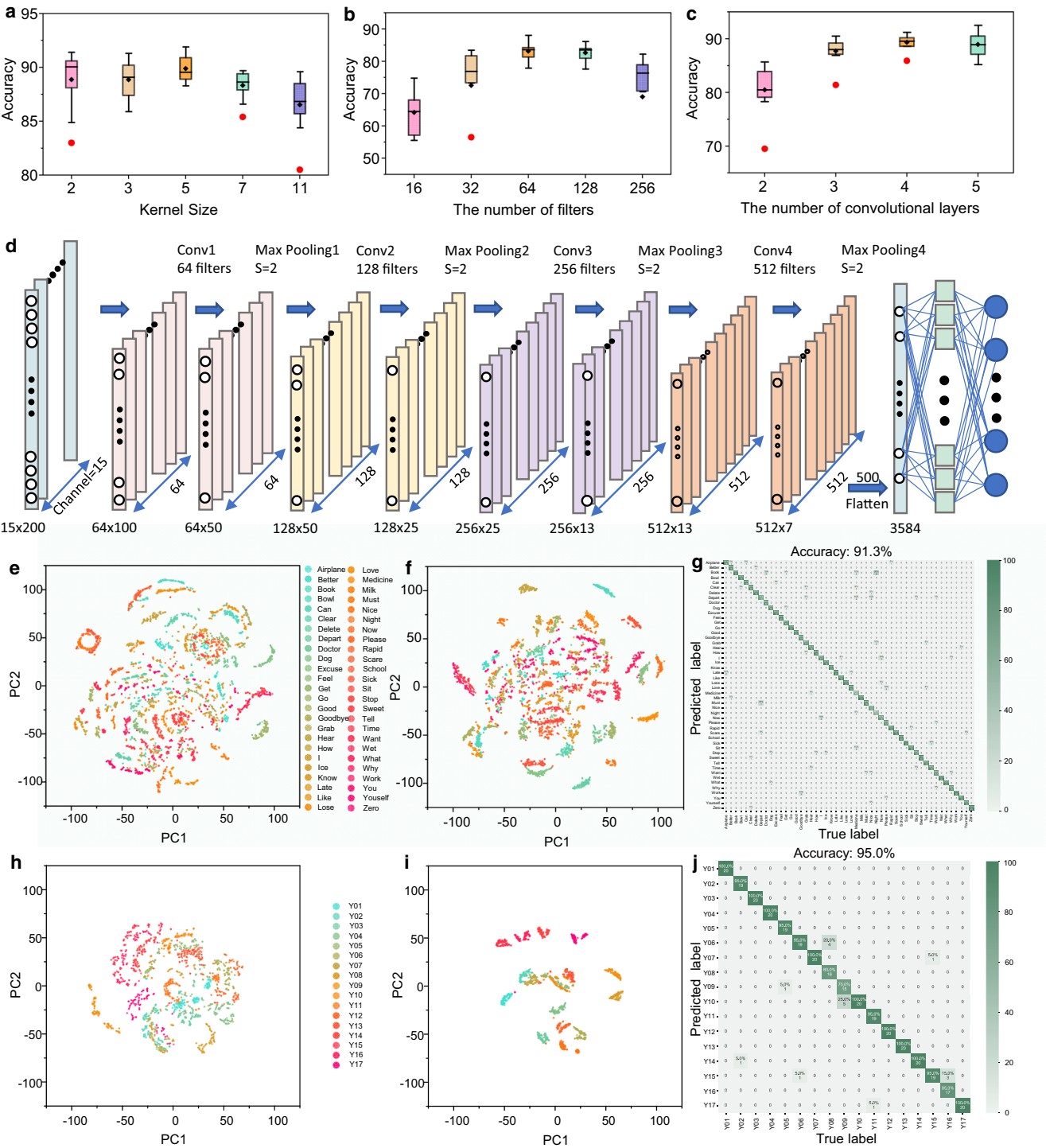

**Fig. 3 Word and sentence recognition based on the nonsegmentation method. a–c** Optimization of CNN structure parameters based on accuracy performance, including Kernel size, the number of filters, and the number of convolutional layers. Boxplots indicate median (middle line), 25th, 75th percentile (box) and 5th and 95th percentile (whiskers) as well as outliers (single points). **d** Final structure of CNN after optimization. **e–f** Cluster results of word signal from CNN input layer and output layer. **g** Confusion map of recognizing 50 words. **h–i** Cluster results of sentence signal from CNN input layer and output layer. **j** Confusion map of recognizing 17 sentences. Source data[65] of Figs. 3g, j are provided in Harvard Dataverse.

sentence consists of the same word elements but in a different order. This is limited by the principal methodology where each word or sentence is labeled as independent distinct items. There is no built-up relation between word units and sentences. Besides, the time latency for recognition is introduced because the CNN model with nonsegmentation method needs to leverage the whole long-data-length signal (generally 200 data points for the word signal and 800 data points for the sentence signal) to trigger the recognition procedure, which is not desirable for the real-time sign language translation. Looking at it in another way, such long-length data comprises substantial features that in turn contribute to a high accuracy of 95% for 17 sentences.

**Word and sentence recognition upon segmentation method**. To explore the feasibility of recognizing new/never-seen sentences, we propose the means of segmentation where signals of all sentences are divided into fragments by the data sliding window, including intact word signals, incomplete word signals, and background signals. These fragments split from all sentence signals are recognized first, and then the CNN model will inversely reconstruct and recognize the sentences. In this way, the segmentation approach enables identifying sentences in the dataset. It also creates the inspiration of recognizing new/never-seen sentence that is composed of word elements that are already recognized before but in a new order. In addition, the recognition latency is significantly reduced due to small-distance sliding (50 data points per sliding) as the recognition process will be triggered with each sliding. In other words, the recognition process is in a simultaneous run when the raw data of sentence inputs, instead of triggering the identification procedure until receiving the entire input as employed in the non-segmentation method. Furthermore, the segmentation approach coupled with deep learning technology provides a universal platform for recognizing new/never-seen sentences to expand the sentence database in a labor-saving manner, eliminating the labor-intensive new sentence data collection.

As shown in Fig. 4a, for the following segmentation of sentence signals, the total 19 words (W01–W19) presented in the 20 sentences of dataset are numbered from 0 to 18 in terms of usage frequency reduction trend in sentences. The word frequency information is shown in Supplementary Fig. 4. The label action is a critical step for the supervised learning of the CNN model. Thus, Fig. 4b demonstrates the detailed process of sentence signal division and labeling by taking the sentence 'The dog scared me' as an example. We introduce a sliding window with a length of 200 data points and a sliding step 50 data points. Considering sample of each sentence is in a fixed data length of 800 points, the sliding window will piece the whole signal into 13 elements, including intact single-word signals, background signals, and mixture signals (i.e., incomplete word signals and background signals). When the sliding window contains the intact word signal, then this fragment signal will be labeled with the number of the corresponding word as listed in Fig. 4a. The background noise is all tagged as 'empty' (W20) with number 19 to denote. Besides, the label of the mixture signal fragment is determined by the principal component. In other words, either word signal or background noise accounts for more than 50% of the sliding window size, and the label will be the number of words or 'empty' 19 correspondingly. The final labeled number series of one data sample for sentence 'The dog scared me' is '[5 5 19 19 19 15 15 15 19 19 0 0 0]' as depicted at the top of Fig. 4b. The schematic segmentation diagram of other three representative sentences, including 'Do you like bowling', 'You need a doctor', and 'I feel better now', are shown in Supplementary Fig. 5a–c, respectively. The table in Fig. 4c summarizes the essential information of investigated 20 sentences: sentence category codes (Y01–Y17 and New1–New3), interpreted sentences, and comprised word/gesture components with different color marks, one exampled label series, and unique number orders. It should be noted that the last three sentences in dark green are selected as new/never-seen sentences, which do not exist in the sentence dataset and are not seen or learned by the CNN model before and are used to verify the potential of recognizing new sentences. Thus, the label series of these three sentences is not visual for the CNN model, and the label status is N.A.

Subsequently, an intuitive and general single classifier is first built to identify all the fragments from 17 sentences without preliminarily filtering empty signals (Fig. 4d). The dataset contains 50 samples for each sentence and hence 850 (50*17) sentence samples in total. The word fragments are from

510 samples for training (60%), 170 samples for validation (20%), and 170 samples for testing (20%). The confusion matrix in Fig. 4e shows an accuracy of 81.9% for the single classifier recognizing word units W01–W20. Afterward, the evaluation on recognizing sentences' performance is our major concern upon inversely reconstructing the whole sentence after classifying all word signal pieces from 17 sentences. Thus, as indicated in Fig. 4f, 10 validation samples (sequence number 1–10) of each sentence are employed to verify the basic capability of reconstructing and recognizing the sentences with the single classifier. The result mapping of sentence recognition illustrates the true and false situation, showing an insufficient average correct rate of 79.41%. For example, the 3rd and 5–10th samples of sentence Y01 are wrongly classified with a poor correct rate of 30%. The nonideal consequence for other sentences, such as Y03, Y04, and Y16, is observed as well. Suffering from a large amount of random and irregular empty signals, the single classifier may not be effective enough when reconstructing and recognizing sentences that existed in the dataset, which may hinder the identification of new/never-seen sentences. To obtain better performance of sentence recognition and then pave the way for discerning new sentences, a classifier with hierarchy architecture is developed as shown in Fig. 4g. The first-level classifier will separate the empty signals and intact word signals to mitigate the negative effect of capricious empty signals on the next-stage recognition of the word signal pieces. Then the word fragments flow to the second-level classifier for precise identification. By this means, the recognition accuracy for word pieces is enhanced to 82.81% as the confusion map in Fig. 4h shows. Eventually, it positively contributes to the sentence recognition, which is indicated by the improved average correct rate of 85.58% with a reduced false area in Fig. 4i. Overall, the hierarchy classifier improves both the accuracy of recognizing word elements and that of identifying ever-seen/trained sentences.

**Recognizing new sentences upon segmentation approach**. Taking the advantages of segmentation, the never-seen sentences New1–New3 are successfully recognized where these sentences are created by new-order word recombination and the order is different from that of sentences in dataset. The process in Fig. 4j demonstrates the recognition of the new/never-seen sentence with the stage of segmentation, real-time sequential fragment identification, and further sentence recognition or translation. Five samples for each new sentence are engaged in this procedure to validate CNN classifiers' feasibility in recognizing new sentences. As provided in Supplementary Table 2, both classifiers render the label series prediction for the total 15 inputs of these 3 new sentences, although the CNN model never sees or learns the true label series before. As expected, the hierarchy classifier shows a reliable performance for new/never-seen sentence recognition, indicated by the less wrong predictions area marked in red. Moving forward to the translation (i.e., recognition) stage (see Supplementary Fig. 6a), the single classifier correctly predicts 2 samples of New1, 3 samples of New2, 4 samples of New3 (average correct rate of 60%). In comparison, the hierarchy classifier achieves precise identification of 5 samples of New1, 3 samples of New2, and 5 samples of New3. In other words, the hierarchy classifier achieves an average correct rate of 86.67% (see Supplementary Fig. 6b), superior to 60% of the single classifier.

Regarding the pros and cons of nonsegmentation and segmentation methods, a radar comparison is provided in Supplementary Fig. 7. Overall, the nonsegmentation approach possesses better performance in the aspect of recognition accuracy either for words or sentences but with the shortcoming of apparent recognition time latency and incapability of new sentence recognition. While the

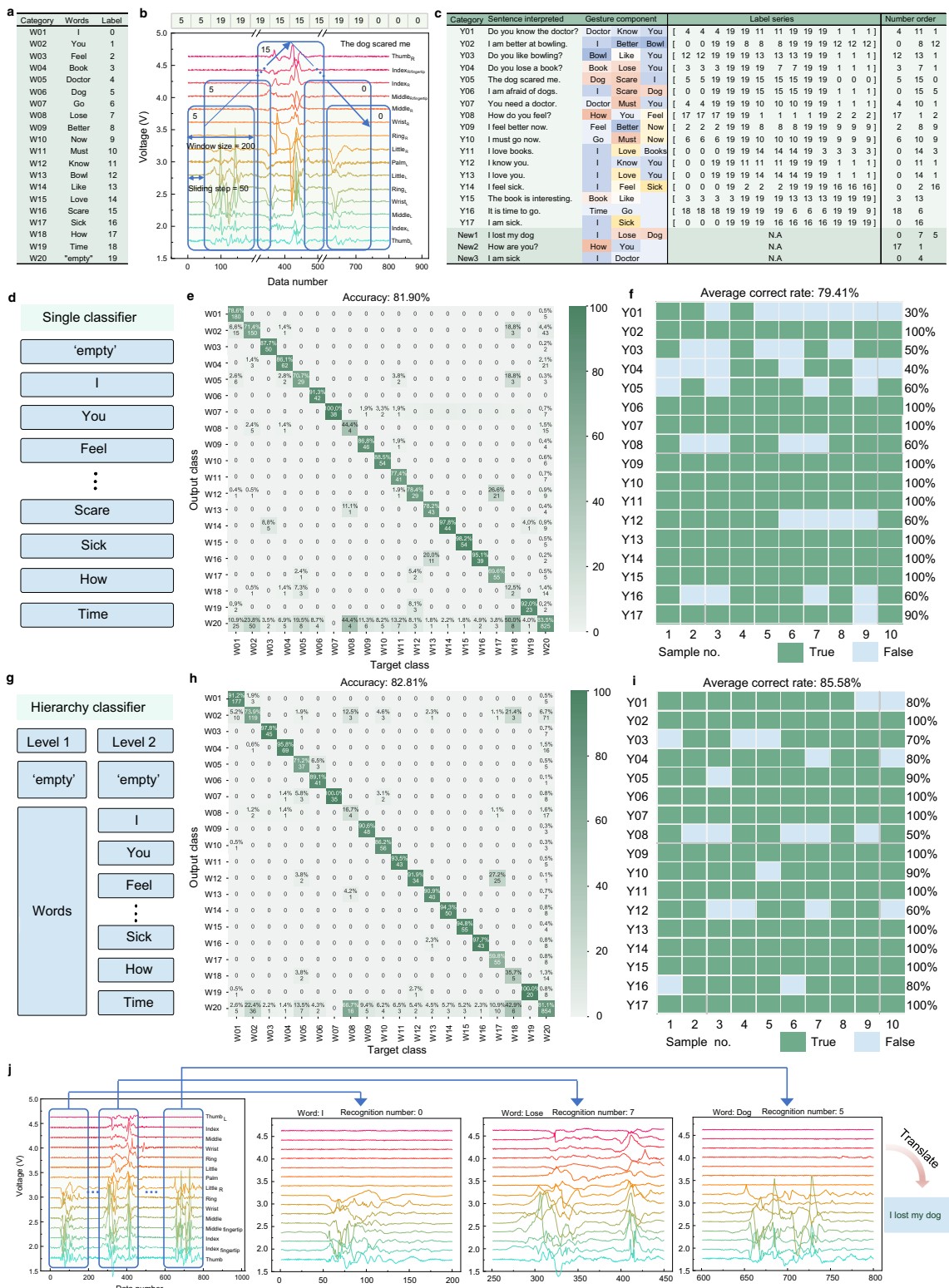

segmentation-assisted CNN model renders significant never-seen/ new sentence identification via the established connection between word units and sentences. Besides, the small-distance sliding induced recognition brings the potential of the real-time recognition or translation of sentences. In addition, compromised recognition accuracy could be overcome by the improved algorithm in the future. Supplementary Note 2 shows a more detailed discussion about the advantages and disadvantages of these two methods.

Besides, Supplementary Note 3 and Supplementary Fig. 8 elucidate the comparison between the most popular image-based gesture recognition and sensor-based solution.

**VR communication interface for the signer and nonsigner.** Most current sign language translation systems simply display the recognition result on visualized interfaces such as mobile phones

**Fig. 4 Word and sentence recognition based on segmentation method, which brings the feasibility of new/never-seen sentence recognition. a** Label table (W01–W20) of 19 words (they are among the total 50 words) that present in 20 sentences (using Y1–Y17 and New1–New3 denote). **b** Schematic diagram of sentence signal segmentation, 'The dog scared me' is taken as an example. **c** Summary table of sentences with category remarks, comprised words, label series, and unique labeled number order. The same word using the same color to mark. **d** Schematic diagram of single classifier. **e** Confusion map of split word element recognition (accuracy 81.9%) based on single classifier. **f** With successful recognition of each element in sentences, the sentences can be inversely reconstructed and recognized at an average correct rate of 79.41% within single classifier. The dark green means right recognition and light blue means wrong prediction. **g** Schematic diagram of the hierarchy classifier. **h** Confusion map of segmented word element recognition (accuracy 82.8%) based on hierarchy classifer. **i** With successful recognition of each element in sentences, the sentence can be inversely reconstructed and recognition at an average correct rate of 85.58% within hierarchy classifier. **j** Recognition process of three new sentences that the CNN model did not learn before, taking 'I lost my dog' as an example. The detailed recognition results can be found in Supplementary Table 2. Source data[65] of Figs. 4e, h are provided in Harvard Dataverse.

and portable displays. However, the acute demand of effective two-way communication between the signers and nonsigners is neglected since most of works do not allow the interaction between signers and nonsigners. Hence, the two-way interaction capability of user interface is in urgent need for the practical application of sign language translation system. The emerging VR interface has been recently employed in various applications with benefiting from enhanced interaction and immersive experience, which would be an ideal platform for an advanced sign language interpretation, visualization, and communication interface. Moreover, the integration of deep learning with VR interface brings a broad prospect for building the intelligent social network, which is targeted for more diversified population groups such as the speech/hearing impaired. Therefore, we develop a deep learning integrated VR interface to realize the bidirectional communication and hope to help the speech/hearing disordered integrate into the majority and enhance the sense of social participation. To meet the requirement of high recognition accuracy in practical applications, the nonsegmented deep learning model is used to achieve the communication in the virtual space. Eventually, the recognition and communication system (Fig. 5a) is comprised of five major blocks including triboelectric gloves for hand motion capture, the printed circuit board (PCB) for signal preprocessing, IoT module Arduino connected PC for data collection, deep-learning-based analytics for signal recognition, and VR interface in Unity for interaction. The recognition results will be projected into cyberspace, in which AI will send corresponding commands based on recognition results and deal with inputs from the nonsigner, to control the communication in VR interface based on Transmission Control Protocol/Internet Protocol (TCP/IP).

In detail, the deep learning block recognizes the nonmainstream sign language, translates it into the prevalent conversational medium such as text and audio, and transmits such information into the next virtual space. As shown in Fig. 5b, the VR interface that is similar to social software is designed for the communication between the speech impaired and healthy user, with switched host–guest views in cyberspace. The client and server are built in the VR interface and linked by TCP/IP, and are accessible for the signer and nonsigner, respectively. Owing to the assistance of deep learning enabled sentence recognition and translation, the client of VR interface allows the speech impaired to use the sign language that they are familiar with to engage in the communication. More precisely, the sign language delivered by the speech impaired is recognized and translated into speech and text by deep learning. Then the speech and text are captured and sent to the nonsigner-controlled server. Next, the nonsigner types directly to respond to the speech-disordered user.

As Fig. 5b(i–v) demonstrates, a greeting scenario is created to demonstrate the feasible communication between the speech/hearing impaired and healthy user under the identical local area network (LAN). Two virtual characters in the VR space, Lily and Mary, represent the created avatars for the signer and nonsigner, with controllable and programmable multiple degrees-of-freedom motions. In the first step Fig. 5b(i), the speech-impaired user Lily performs sign language 'How are you?', which is recognized and converted to text and audio by the deep learning model. By means of TCP/IP, the client connected with deep learning component receives the text/speech message 'How are you?' and transmit it to the nonsigner Mary controlled server. Projecting to the virtual space, the signer avatar Lily slightly lifts her hand to greet her friend (i.e., the nonsigner Mary). Correspondingly, the nonsigner types 'Not good. I have a stomachache.' to respond to the signer. The virtual girl Mary represents the nonsigner shakes her head and covers her stomach with hands to show her illness, as shown in Fig. 5b(ii). Then the speech-impaired user replies to the nonsigner Mary with the sign language 'You need a doctor' (Fig. 5b(iii)). Figure 5b(iv) and Fig. 5b(v) similarly show their interaction. The detailed conversation demonstration can be found in Supplementary Movie 3. The conversation of Fig. 5b is summarized in Fig. 5c. The VR communication interface linked with AI allows interaction of the speech/hearing disordered and healthy people closely and even remotely, providing a promising platform for the immense interactions between two population groups.

## Discussion

Sign language recognition and translation are of great significance to remove the communication barrier between the speech/hearing impaired and the general public. Nowadays, wearable HMIs are emerging as an innovative assistive platform to implement direct, conformable hand motion measurement. As a paradigm for the wearable HMI, gloves have received intense research efforts for sign language recognition due to its capability to seamlessly detect the delicate motions of our dexterous hands. However, current glove solutions are generally limited to identify discrete and simple numbers, letters, or words of sign language by relying on signal magnitude. Although few works have employed AI-enabled advanced data analytics to achieve highly precise sign language translation, they are still limited to the classification of discrete words that cannot meet the requirement of practical communication in the signer's daily lives. The lack of effective approach to perceive sentences hinders the communication of the signer with nonsigner (see Supplementary Table 3).

To realize advanced, comprehensive, and practical sign language recognition, we herein propose a sign language recognition and communication system comprising a smart triboelectric glove, AI block, and the back-end VR interface. The system enables the separate and independent recognition of words (i.e., single gestures) and sentences (i.e., continuous multiple gestures) with high accuracies of 91.3% and 95% within nonsegmentation frame. Furthermore, to overcome the limitation of incapability of recognizing new/never-seen sentences, the segmentation method is proposed. It divides all the sentence signals into word fragments while AI learns

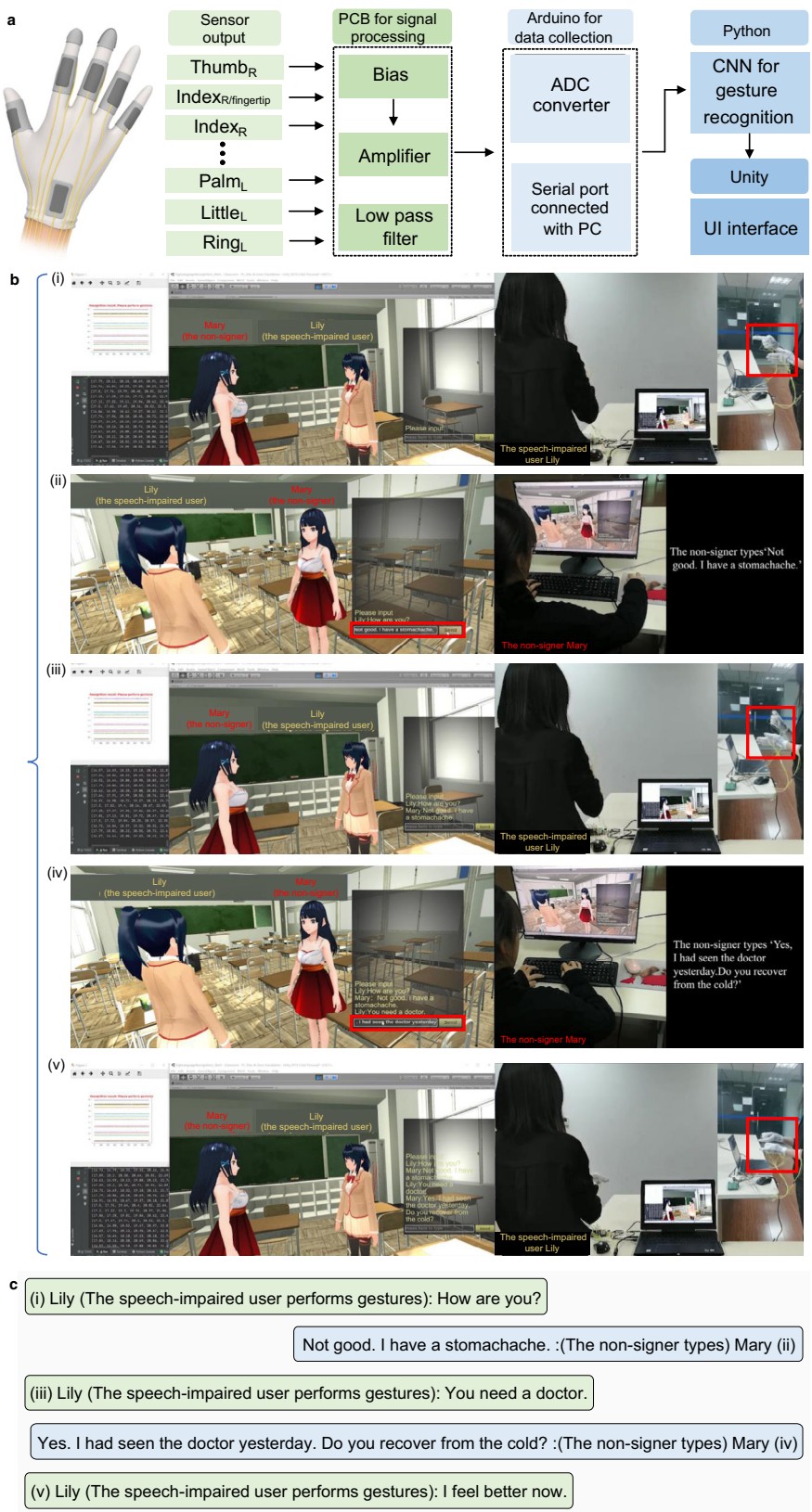

c

(i) Lily (The speech-impaired user performs gestures): How are you?

Not good. I have a stomachache. :(The non-signer types) Mary (ii)

(iii) Lily (The speech-impaired user performs gestures): You need a doctor.

Yes. I had seen the doctor yesterday. Do you recover from the cold? :(The non-signer types) Mary (iv)

(v) Lily (The speech-impaired user performs gestures): I feel better now.

and memorizes all split elements. Then the deep learning architectures, single and hierarchy classifiers, inversely infer, reconstruct and recognize the whole sentence (recognition accuracy: 85.58%) benefitting from the established correlation of basic word units and sentences. Furthermore, the segmentation approach renders new/never-seen sentence recognition, in which the new/never-seen

sentences are not included in the dataset and created by recombining the learned word units in new orders. Finally, the embedded VR interface can act as the bridge of user terminals, transmitting messages back and forth, configuring a closed-loop communication system with gloves and AI component. On the VR platform, the speech/hearing impaired can directly perform sign language to

**Fig. 5 The demonstration of communication between the speech impaired and the nonsigner. a** Flow chart of the sign language recognition and communication system, which allows the signer to use sign language and nonsigner to types directly to engage in the interaction. The delivered sign language by the signer is recognized and translated into text and speech by AI block. Based on TCP/IP, the client (controlled by signer Lily) in VR interface receives the recognition results and transmits to the sever (operated by the nonsigner Mary). The nonsigner types on the chat box to respond to the signer. **b** (i–v) Communication/conversation process in VR interface between the speech-disordered user Lily and nonsigner Mary based on the sign language recognition and communication system. The red rectangle indicates the corresponding reaction of these two users. These photos are of one of the authors. **c** Conversation summary of **b**. The hand image is created by the authors via Blender. Photo credit: Feng Wen, National University of Singapore.

interact with the nonsigner (i.e., human-to-human interaction), while the nonsigner directly involves in the communication process via direct typing. In summary, we have successfully achieved recognition of 50 words and 20 sentences picked up from the American sign language phrase book as demonstration, which can be further broadened by leveraging more sentences in the dataset to meet the practical daily communication demand. Hopefully, the segmentation method provides a universal and practical platform to expand the sentence database in practical communication by new sentence inclusion, largely improving the sign language translation system's practicality. Besides, the intelligent system consist of smart gloves, AI-assisted recognition, and TCP/IP enabled server-client LAN communication in virtual space, providing a reliable and appealing platform for the seamless and immense interactions between nonsigners and signers either remotely or proximately.

## Methods

**TENG sensor fabrication.** The 00-30 Ecoflex with a 1:1 weight ratio of part A and part B is coated on the conductive textile (for electrode) for consolidation. Then the positive wrinkled nitrile attached on the conductive textile contacts with the negative Ecoflex layer to assemble the triboelectric sensor. Finally, the conductive electrodes are encapsulated by nonconductive textiles to shield the ambient electrostatic effect. Different sensors with different areas are fabrication for customized glove configuration.

**Characterization of sensor triboelectric output.** The voltage measurements are carried out by oscilloscope (Agilent, InfiniiVision, DSO-X 3034 A) with the normal probe 10 MΩ.

**Glove configuration.** The as-prepared two-layered triboelectric sensor is sewed into the small cotton textile pocket. The finalized encapsulated triboelectric sensors for each hand position are stitched on the glove by E7000 textile glue for a seamless fit with fingers.

**Data collection and dataset configuration.** The signal acquisition module acquires the triboelectric signals generated by different gestures in Arduino MEGA 2560 microcontroller. For the nonsegmentation method, 100 samples for each word and sentence are collected where 80 samples (80%) for training and 20 samples (20%) for testing. Thus, there are 5000 samples for total 50 words and 2000 samples for total 20 sentences. For the segmentation approach, since all the data from sentences, there are 50 samples with number series labels for each sentence (17 sentences and hence 850 samples in total). The division window in the Matlab workspace with the length of 200 data points slides at a step of 50 data points to extract the signal fragments. Thus, each sentence sample with 800 data points is segmented into 13 elements. The extracted data elements are labeled with numbers of corresponding words or 'empty'. Each number represents a signal fragment that may be an intact gesture signal, background noise, or an incomplete gesture signal. The 60% samples of each number (i.e., 0–19) are used for training, 20% for validation, and 20% for testing. Finally, 5 samples for 3 new/never-sentences are employed to verify the capability of new sentence recognition of the segmentation-assisted CNN model without prior training.

**The detailed mathematics behind PCA.** PCA is used to reduce the dimensions while maximum retaining the information. The principle of PCA relies on the correlation between each dimension and provides a minimum number of variables, which keeps the maximum amount of variation about the distribution of original data. PCA employs the eigenvalues and eigenvectors of the data-matrix, which can point along the major variation directions of data to achieve the purpose of dimension reduction. For the detailed mathematics of PCA[59], the vector X includes the component that is the input signal $x_i$ for each gesture,

$$X = \{x_1, x_2, x_3, \dots, x_n\} \tag{1}$$

Then the mean value $X_{mean}$ of X is calculated as,

$$X_{mean} = \frac{1}{n} \sum_{i=1}^{n} x_i \tag{2}$$

Determining the difference $d_i$ between the input and mean value,

$$d_i = x_i - X_{mean} \tag{3}$$

Based on Eq. (3), we get the covariance matrix S,

$$S = \frac{1}{n} \sum_{i=1}^{n} d_i d_i^T \tag{4}$$

According to linear algebra, the eigenvalues $\lambda$ and eigenvectors p of the covariance matrix are defined as,

$$Sp = \lambda p \tag{5}$$

Because there are more than one the eigenvalues and eigenvectors, the principal component matrix P is,

$$P = \{p_1, p_2, p_3, \dots p_k\} \tag{6}$$

Thus, the input signal can be projected into a new output matrix Y with reduced dimensions,

$$Y = \{y_1, y_2, y_3, \dots, y_i, \dots, y_n\} \tag{7}$$

where $y_i = P^T(x_i - x_{mean})$ and k controls the principal component matrix P and hence controls the dimension of output Y.

**Demonstration of pure words and sentences recognition.** For word and sentence recognition demonstration in Fig. 3, the triboelectric glove with 15 sensor channels is connected to the Arduino MEGA 2560. By serial communication with Python in PC, the raw data of different gestures is acquired and recognized by the trained CNN model with Tensorflow and Keras frames, then displaying recognition results on the Python interface.

**VR communication interface building and demonstration.** On top of details described in the last section 'Demonstration of pure words and sentences recognition', the recognized result in Python is sent into virtual Unity space via TCP/IP communication and displayed on the VR interface. The server (for nonsigners) and client (for signers) terminals building in Unity also rely on TCP/IP. Within the identical LAN, the speech/hearing impaired and healthy people could remotely communicate via the VR interface where the speech/hearing disordered uses their familiar sign language, and the nonsigner types directly. The authors affirm that the participant provide informed consent for publication of the images in Figs. 2a, 5b, Supplementary Figs. 2a, 8a, and the videos in Supplementary Movies 1, 2, 3.

## Data availability

The source data of Figs. 2c, e, 3g, j, 4e, h, and the dataset of machine learning that support the findings of this study are available in Harvard Dataverse, https://doi.org/10.7910/DVN/7KJWV3. All other relevant data are available from the corresponding author upon reasonable request.

## Code availability

The codes that support the findings of this study are available from the corresponding authors upon reasonable request.

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

## Acknowledgements

This work by the authors is partially/fully supported by the National Key Research and Development Program of China (Grant No. 2019YFB2004800, Project No. R-2020-S-002) at NUSRI, Suzhou, China; the Advanced Research and Technology Innovation

Centre (ARTIC), the National University of Singapore under Grant (project number: R261-518-00x-720); the National Research Foundation, Singapore under its AI Singapore Programme (AISG Award No: AISG-GC-2019-002); the "Smart sensors and artificial intelligence (AI) for health" seed grant (R-263-501-017-133) at NUS Institute for Health Innovation & Technology (NUS iHealthtech). Any opinions, findings and conclusions or recommendations expressed in this material are those of the author(s) and do not reflect the views of National Research Foundation, Singapore.

## Author contributions

F.W. and C.L. conceived the idea. F.W. planned and performed the experiments. F.W. took all the photos shown in figures. F.W. and Z.Z. wrote the deep learning algorithms and control programs for demonstration. F.W., Z.Z., T.H., and C.L. contributed to the data analysis and drafted the manuscript. F.W., Z.Z., T.H., and C.L. edited the manuscript. C.L. supervised the work.

## Competing interests

The authors declare the following competing interests: F.W., Z.Z., T.H., and C.L. are inventors on the patent application (pending, Ref: 2021-187, serial no. 10202109596U) submitted by National University of Singapore that covers sign language recognition and communication system.
