## [Peer Review File · Nature Communications]

AI enabled sign language recognition and VR space
bidirectional communication using triboelectric smart gloveREVIEWER COMMENTS

Reviewer #1 (Remarks to the Author):

This paper introduces a CNN and TENG based smart glove. The smart glove is ingeniously designed and powerful. Especially, the combination with VR is innovative, shows its interesting application prospects. The manuscript has clear logic, complete structure and detailed data. Meanwhile, this manuscript needs some revise. For example, some statements in this manuscript are not clear, and some pictures needs more explain. It is recommended to accept after a minor revision.

1. Feature engineering is a very important part of artificial intelligence. Please explain in detail some important features extracted from raw data in this work. And the mathematical methods to get these features.
2. The sensor and actual signal output of some gesture calls in Figure 2b and S2 seem to be inconsistent. Please explain the reason for this situation.
3. There are many similar jobs combining gloves and machine learning. It is recommended to add some comparison for this work with other works to reflect the significant advantages of this job.
4. Response time is an important parameters for real-time system. How long does it take for the entire system to process an action and complete the voice broadcast? How long is spent on hardware? And how long on software?
5. The article has some language problems and typos, check the full text carefully, and make corrections. For example, in line 303, the word 'irideal' seems incorrect.

Reviewer #2 (Remarks to the Author):

The authors propose a practical sign language recognition and communication system using triboelectric glove sensors integrated with artificial intelligence (AI) data analytics, and virtual reality (VR) based mutual communication interface. The sign language recognition and communication system enables the identification of diversified words and sentences. In particular, this system has the capability of recognizing new sentences via sentence signal splitting and new-order pieces recombination within segmentation method. The segmentation method provides a universal solution to readily expand sentence database through new sentences recognition. Besides, the back-end VR interface allows proximate or remote two-way communication between the signers and non-signers instead of

unidirectionally delivering sign language recognition results, creating the feasibility of practical application of sign language recognition system. With the above information, I recommend minor revision with the next few points:

1. Why is the palm sensor located on the left hand? What is the reason behind this arrangement?
2. As Figure 1(b) shows, some gestures in sign language involve elbow/shoulder motions even facial expression. It may be an obstacle for detecting more gestures by only using glove. May the authors illustrate how to make more gestures detectable and enrich the recognized sign language database?
3. Since the statistical analysis is employed to indicate the necessity of introducing advanced deep learning algorithm, the related description could be divided into a separated section from 'Glove configuration and sensor characterization'. It may make the manuscript frame more coherent.
4. The main text just mentions principal component analysis (PCA) in line 221, in which extracted features are high-dimensional data. What is the dimension reduction and visualization method of Figure 3(e), (f), (h), and (i)?
5. Compared with Nat. Electron., 3(9), 571-578 (2020) which is mentioned as Ref. 59 in your manuscript, may the authors emphasize the advantages of your work in comparison?
6. In section 'Recognizing new/never-seen sentences upon segmentation approach', line 329 shows the hierarchy classifier achieves an average correct rate of 86.67% for recognizing new sentences. Although the authors provide the detailed Table S2, it does not directly indicate the average correct rate 86.67%. The authors may add one graph like Figure 4(f) and (i) to clearly show the recognition accuracy of new sentences.
7. How would the authors evaluate the two approaches mentioned in this work, i.e., the segmentation and non-segmentation methods?
8. For the VR demonstration in Figure 8, why the signer cannot directly type in the message to communicate with the healthy subject? Since there is a laptop or computer, what would be the necessity of the machine-learning-enabled glove? How is it beneficial for the signers in practical applications?
9. As the authors mention, there are few publications showing gesture sensing and even sign language recognition based on resistive, capacitive, and triboelectric effects. Recognition of sentences even new sentences of sign language using triboelectric glove sensors with AI integration could bring great significance to the research field. The authors could provide a benchmarking table in Supplementary Materials to summarize the features of various reported works in comparison to current work in terms of function, applications etc. It helps show the uniqueness of current work in an intuitive way.

Reviewer #3 (Remarks to the Author):

In this manuscript, the authors propose a sign language recognition and communication system, including gloves integrated with triboelectric sensors, AI block and VR interactive interfaces. AI analysis of TENG sensor data is very important for TENG and AI application communities. I recommend the manuscript to publish after major revision. There are several comments as the following:

1. There are two main sign language recognition approaches: image-based and sensor-based. [IEEE transactions on human-machine systems, 2014, 44(4): 551-557] For image-based systems, signers use simple devices. For sensor-based systems, smart gloves need personalized design, which are expensive. TENG can overcome the challenge of cost. Compared with image-based technology, authors should provide experiments and discuss the advantages of TENG sensor-based systems.
2. Traditional commercial sensors are good candidate to obtain high-precision data. The author should compare TENG sensors with traditional commercial sensors? How to redesign AI and communication system to overcome the limitation of TENG, for example, accuracy and the principle of orthogonality of the sensor signal? The authors should add experiments to provide the important properties of sensors.
3. AI technology needs to recover information and improve the accuracy of information by learning the characteristics of complete information. [Nano Energy, 2021: 105887] Especially, the data obtained by the TENG sensor is random and low accuracy. The authors should provide the detailed discussion of TENG sensor-based systems.

Dear Reviewers:

Your comments are all valuable and very helpful for revising and improving our manuscript. We have studied all of the comments carefully and made the revisions which we hope can meet with your approval. Briefly speaking, we have done extra experiments to collect new data and added the additional discussion into the main manuscript and supplementary in response to reviewers' comments. We have revised Figure 2, Figure 4, Figure 5, and Figure S2 and added Figure S6-Figure S8, Note 1-Note 3, and Table S3 in the Supplementary Materials. Besides, Figure R1-Figure R3 is included in the response letter. All of the revisions in the main manuscript and the supplementary information are marked in red, and the point-by-point response to the reviewers' comments are provided as follows:

REVIEWER COMMENTS

Reviewer #1 (Remarks to the Author):

This paper introduces a CNN and TENG based smart glove. The smart glove is ingeniously designed and powerful. Especially, the combination with VR is innovative, shows its interesting application prospects. The manuscript has clear logic, complete structure and detailed data. Meanwhile, this manuscript needs some revise. For example, some statements in this manuscript are not clear, and some pictures needs more explain. It is recommended to accept after a minor revision.

Thanks a lot for the reviewer's comments. The following is our point-to-point response.

1. Feature engineering is a very important part of artificial intelligence. Please explain in detail some important features extracted from raw data in this work. And the mathematical methods to get these features.

We thank the reviewer's valuable comment. As shown in Figure 3(d), we use five convolutional layers and five max pooling layers for feature extraction. To better understand the clustering performance of the proposed CNN structure, we extract the results of the last fully-connected layer (each data sample with 3000 points finally has been stretched to 3584 data points after convolution and pooling) to compare with the raw data. Regarding the feature clustering result in Figure 3, due to the high dimensionality of extracted data, we first use principal component analysis (PCA) to reduce the dimensions whilst maximum retaining the information. The principle of PCA relies on the correlation between each dimension and provides a minimum number of variables which keeps the maximum amount of variation about the distribution of original data. PCA uses the eigenvalues and eigenvectors of the data-matrix, which can point along the major variation directions of data. For the detailed mathematics of PCA^[1], the vector X includes the component that is input signal x_i for each gesture,

$$X = \{x_1, x_2, x_3, \dots, x_n\} \quad (1)$$

Then the mean value X_{mean} of X is calculated as,

$$X_{mean} = \frac{1}{n} \sum_{i=1}^n x_i \quad (2)$$

Determining the difference d_i between the input and mean value,

$$d_i = x_i - X_{mean} \quad (3)$$

Based on (3), we get the covariance matrix S ,

$$S = \frac{1}{n} \sum_{i=1}^n d_i d_i^T \quad (4)$$

According to linear algebra, the eigenvalues λ and eigenvectors p of the covariance matrix is defined as,

$$Sp = \lambda p \quad (5)$$

Because there are more than one the eigenvalues and eigenvectors, the principal component matrix P is,

$$P = \{p_1, p_2, p_3, \dots, p_k\} \quad (6)$$

Thus, the input signal can be projected into a new output matrix Y with reduced dimensions,

$$Y = \{y_1, y_2, y_3, \dots, y_i, \dots, y_n\} \quad (7)$$

Where $y_i = P^T(x_i - x_{mean})$ and k controls the principal component matrix P and hence controls the dimension of output Y .

On the basis of PCA, we reduced the data of the input layer and the last fully-connected layer to 30 dimensionalities. After PCA, the t-distributed Stochastic Neighbor Embedding (t-SNE) is used for the visualization of the dataset. The t-SNE can evaluate the original data that is entered into the algorithm and find the best way of representing the data using fewer dimensions (e.g., two dimensions) by matching both distributions. The t-SNE result of the last fully-connected layer is shown in Figure 3(f) and Figure 3(i), which acquires much clearer boundaries of different classes indicating the desirable performance of proposed CNN methods. The discussion of PCA is added in Page 8 and Page 18 of the manuscript while the description of t-SNE for visualization is included in Page 9 of manuscript for your evaluation.

Figure 3. Word and sentence recognition based on the non-segmentation method. (a)-(c) The optimization of CNN structure parameters based on accuracy performance, including Kernel size (a), the number of filters (b), and the number of convolutional layers (c). (d) The final structure of CNN after optimization. The cluster results of word signal from CNN input layer (e) and output layer (f). (g) The confusion map of recognizing 50 words. The cluster results of sentence signal from CNN input layer (h) and output layer (i). (j) The confusion map of recognizing 17 sentences.

2. The sensor and actual signal output of some gesture calls in Figure 2b and S2 seem to be inconsistent. Please explain the reason for this situation.

We thank the reviewer for the careful review and comment. The incident inconsistency in Figure 2 and Figure S2 has been revised. We also have thoroughly checked the total 50 gestures and their corresponding signals shown in these two figures to avoid such mistakes. Besides, the existed crosstalk of 15 glove sensors can

lead to an imperfect match between the gesture and triboelectric signals. Please refer to Page 25 of the manuscript and Page 5 of Supplementary Materials for the view of revised Figure 2 and Figure S2.

3. There are many similar jobs combining gloves and machine learning. It is recommended to add some comparison for this work with other works to reflect the significant advantages of this job.

We appreciate the reviewer's constructive comment. The relevant works for glove-based sign language recognition are summarized in the following table. **Above all**, most reported sign language recognition based on the glove solution identifies the simple and limited variety of numbers and letters with manual simple feature extraction (e.g., signal amplitude and polarity). Although few works demonstrated the incorporation of artificial intelligence (AI) data analytics to realize the classification of a dozen sign language gestures in high accuracy, it still lacks an effective and practical approach for sentence recognition of sign language, which is of paramount significance to remove the communication barrier between signers and non-signers. By contrast, our AI-enabled sign language recognition system achieves the classification of 50 words and 20 sentences. In particular, through splitting sentence signals into word units and recombining these units in new sequences, the proposed CNN architecture renders never-seen/new sentence recognition. It provides a universal methodology to readily expand the sentence database, significantly improving the sign language recognition system's practicality. **Second**, the interfaces (e.g., mobile phone or PC) that sign language recognition results are projected or displayed usually do not realize the two-way communication between signers and non-signers. In other words, these systems just unidirectionally deliver the translated text or audio to the users. This kind of human-to-human interaction hindrance could greatly limit the practical application of the sign language recognition system. In our work, coupling the interconnected client and server of *Unity* with the front-end AI component, the signers can directly use sign language in the client end to participate in the communication process. While the healthy person types in the server end to respond. The summarized table is included in Page 16 of Supplementary Materials as Table S3 for your evaluation.

Table S3. The benchmarking table for comparing with other similar works

Mechanism	Device	Method	Sensor/hand	Gesture	Sentence	Recognizing new sentence	Self-powered	Mutual interaction	Ref
Piezoresistive	Armband	SVM	3	8 letters	-	-	x	x	1
Piezoresistive	Radar+wristband	HSVM	5	4 letters	0	x	x	x	2
Resistive	Finger sensing skin	SVM	5	10 letters	0	x	x	x	3
Resistive	Glove	BNN	9	10 letters	0	x	x	x	4
Resistive	Elastomer Glove	Amplitude	5	5 numbers	-	-	x	x	5
Resistive	Glove	Amplitude	5	9 words	0	x	x	x	6
Resistive	Glove	Amplitude	9	26 letters	0	x	x	x	7
Capacitive	Glove	Amplitude	5	5 numbers	-	-	x	x	8
Capacitive	Glove	Amplitude	5	36 letters	0	x	x	x	9
Capacitive	Wristband	Amplitude	15	15 words	0	x	x	x	10
Capacitive	Glove	XRF	16	10 numbers	0	x	x	x	11
Capacitive	sensing skin	Amplitude	4	6 numbers	-	-	x	x	12
Ionic	Glove	Amplitude	8	6 words	0	x	v	x	13
Piezoelectric	Glove	Amplitude	5	6 numbers	0	x	v	x	14
Triboelectric	Glove	Amplitude	5	6 numbers	-	-	v	x	15
Triboelectric	Glove	Amplitude	6	7 numbers	-	-	v	x	16
Triboelectric	Glove	Amplitude	5	5 letters	0	x	v	x	17
Triboelectric	Glove	SVM	5	11 letters	0	x	v	x	18
Triboelectric	Glove	CNN	7	50 words	20	v	v	v	*

Note: (H)(SVM): (hierarchical) support vector machine; BNN: binary neural network; XRF: extremely randomized trees; CNN: convolutional neural network. (*This work)

- Response time is an important parameter for real-time system. How long does it take for the entire system to process an action and complete the voice broadcast? How long is spent on hardware? And how long on software?

We thank the reviewer for this constructive comment. The response time could be obtained from Video S3 (The sign language recognition and communication system for interaction of speech-impaired user and non-signer). The schematic diagram of the time axis along with critical events is shown below for an intuitive view of the response time. As shown in Figure R1, it takes around 9.5s for the whole system to collect the triboelectric gesture signal and to achieve ultimate audio delivery. **First**, the duration for data collection in *Arduino* is artificially set on the basis of the consuming time for performing gestures. In our specific case, the maximum time for three sentences present in Video S3 is 4.6s. Thus, we determine the data collection time spent on the hardware *Arduino* as 8s containing reserved time for human reaction, which is indicated by the grey area in Figure R1. **Second**, *Python* extracts the data and accomplishes processing and recognition only within 0.1s owing to the strong computation capability. **Third**, *Unity* responds to the command from *Python* and executes the corresponding actions (e.g., voice delivery). Ideally, the response would be triggered as soon as *Unity* receives the order. In fact, this response process consumes 1.4s, in which such delay derives from *Unity* itself. Accordingly, the software *Python* and *Unity* spend 1.5s in total, as indicated by the light yellow and orange areas. **Overall**, the time spent on hardware for data collection is dominant with accounting for ~84.2% (8/9.5). Luckily, it is flexible and adjustable based on the speed of gesture making. The response time for software only occupies ~15.8% (1.5/9.5), which could be acceptably ignored.

Figure R1. The time axis from data collection to final speech broadcasting.

- The article has some language problems and typos, check the full text carefully, and make corrections. For example, in line 303, the word 'irideal' seems incorrect. Thanks for the careful review and kind reminder. We have revised the typos and thoroughly checked the entire manuscript to avoid this kind of mistake.

Reviewer #2 (Remarks to the Author):

The authors propose a practical sign language recognition and communication system using triboelectric glove sensors integrated with artificial intelligence (AI) data analytics, and virtual reality (VR) based mutual communication interface. The sign language recognition and communication system enables the identification of diversified words and sentences. In particular, this system has the capability of recognizing new sentences via sentence signal splitting and new-order pieces recombination within segmentation method. The segmentation method provides a universal solution to readily expand sentence database through new sentences recognition. Besides, the back-end VR interface allows proximate or remote two-way communication between the signers and non-signers instead of unidirectionally delivering sign language recognition results, creating the feasibility of practical application of sign language recognition system. With the above information, I recommend minor revision with the next few points:

Thanks a lot for the reviewer's comments. The following is our point-to-point response.

1. Why is the palm sensor located on the left hand? What is the reason behind this arrangement?

We appreciate the reviewer's valuable comment, and we are sorry the explanation about the sensor distribution on gloves was insufficient. As depicted in Figure 1(b), the statistical analysis finds that daily sign language involves three major motions, including elbow/shoulder motions, face muscle activities, and hand movements. The dominant hand motion accounts for 43%. Thus, hand motion sensing is inevitable for sign language recognition. As shown in the enlarged pie chart in the right of Figure 1(b), the hand motions can be subdivided into four categories, including finger bending (56%), wrist motion (18%), touch with fingertips (16%), and interaction with palm (10%). These detailed hand motions need sensors in different positions of hands to generate the essential correspondence.

Figure 1(c) and Figure S1 show the triboelectric sensor is mounted on each finger for finger bending detection, while two sensors are put on wrists for wrist motion perception. In addition, the fingertips of the index and middle of the right hand are also in frequent use in daily used sign language, and hence two sensors are located at fingertips. Meanwhile, signers often use their palms to interact with other parts of their bodies to convey richer information. But we nominally allocate only one sensor on the palm of the left hand rather than two sensors, one located on the left hand and one located on the right hand. There are **two major considerations behind such arrangement**: (1) based on the minimalist design for reducing system complexity, and we expect as few sensors as possible with the limit of capable of detecting necessary hand motions. Thus, only one sensor is located on the left hand instead of one for the left hand and one for the right hand. (2) This sensor is attached on the left hand not the right hand. Because the final status of most of gestures that involve palm end up on the palm of the left hand, such as 'Excuse', 'Medicine', 'Nice', 'School', 'Stop' and 'What' shown in Figure S2. Hence one palm sensor on the left hand is reasonable to sense the interaction motions. The above discussion is added in Page

3 of Supplementary Materials as Note S1.

Figure 1. The sign language recognition and communication system. (a) Schematics of glove-based human-machine interface for sign language interpretation. (b) The proportion of different motions that are commonly used in sign language helps determine sensor position on gloves. (c) The sensor position on gloves based on hand motion statistics in (b). Detailed area information of the sensor on each position can be found in Supporting Material Figure S1. (d) Materials of the triboelectric sensor. (e)-(h) Basic characterization of triboelectric sensor units. The voltage output dependence on key parameters including sensor area (e), force (f), bending degree (g), and bending speed (h) is investigated.

2. As Figure 1(b) shows, some gestures in sign language involve elbow/shoulder motions even facial expression. It may be an obstacle for detecting more gestures by only using glove. May the authors illustrate how to make more gestures detectable and enrich the recognized sign language database?

Thanks a lot for the reviewer's valuable comment and suggestion. This is a good point for exploring the potential of a new design of smart gloves in order to sense more gestures and improve the recognition accuracy. Indeed, although the hand motion is more dominant (43%), the elbow motion yet represents 37% of sign language related motions (Figure 1(b)). To achieve more comprehensive and

advanced sign language recognition, the sensing of elbow, even entire arm movements, is significant for the next-step development.

Overall, there are **three possible ways** by referring to others' works: (1) integrated design of sleeve and glove. (2) inertial measurement unit (IMU) to detect dynamic motion of human upper limbs. (3) integrating visual data with sensory data. **In detail**, the sleeve with customized sensors located at the elbow or other proper positions may provide necessary sensory data for upper limb motions, integrated with the smart glove that is able to perform more direct measurements of joint angles and mechanical interactions. ^{[2][3]} **On the other hand**, IMU is a mature technique that could be incorporated with flexible/wearable electronics (e.g., glove) to capture dynamic motions. ^[4] **Finally**, integration of sensor-based and vision-based enables multi-modal gesture sensing, in which the visual component could catch upper-limb motions and even facial expressions. Meanwhile, the sensory glove component offers intimate measurement of hand motions and overcomes the issue of visual images/videos (e.g., susceptible to light conditions or occlusion). It has been proved that recognition accuracy is improved by combing visual data with sensory data from wearable sensors. ^[5] With the assistance of artificial intelligence (AI), the above three integration solutions have great opportunities to achieve the recognition of more diversified gestures toward advanced sign language applications.

3. Since the statistical analysis is employed to indicate the necessity of introducing advanced deep learning algorithm, the related description could be divided into a separated section from 'Glove configuration and sensor characterization. It may make the manuscript frame more coherent.

Thank the reviewer for bringing this to our attention. We have moved the description of statistical analysis to a separated section with the section title 'Statistical analysis of signals of 50 words and 20 sentences. Please refer to line 167, Page 7 of the manuscript.

4. The main text just mentions principal component analysis (PCA) in line 221, in which extracted features are high-dimensional data. What is the dimension reduction and visualization method of Figure 3(e), (f), (h), and (i)?

We thank a lot for the reviewer's careful comment. Regarding the feature clustering result in Figure 3(e), (f), (h), and (i), due to the high dimensionality of the raw signal and the extracted data from the last fully-connected layer, we first use principal component analysis (PCA) to reduce the dimensions while maximum retaining the information. The principle of PCA relies on the correlation between each dimension. It provides a minimum number of variables that keeps the maximum amount of variation about the distribution of original data. PCA uses the eigenvalues and eigenvectors of the data-matrix, pointing along the major variation directions of data. For the detailed mathematics of PCA ^[1], the vector X includes the component that is input signal x_i for each gesture,

$$X = \{x_1, x_2, x_3, \dots, x_n\} \quad (1)$$

Then the mean value X_{mean} of X is calculated as,

$$X_{mean} = \frac{1}{n} \sum_{i=1}^n x_i \quad (2)$$

Determining the difference d_i between the input and mean value,

$$d_i = x_i - X_{mean} \quad (3)$$

Based on (3), we get the covariance matrix S ,

$$S = \frac{1}{n} \sum_{i=1}^n d_i d_i^T \quad (4)$$

According to linear algebra, the eigenvalues λ and eigenvectors p of the covariance matrix is defined as,

$$Sp = \lambda p \quad (5)$$

Because there are more than one the eigenvalues and eigenvectors, the principal component matrix P is,

$$P = \{p_1, p_2, p_3, \dots, p_k\} \quad (6)$$

Thus, the input signal can be projected into a new output matrix Y with reduced dimensions,

$$Y = \{y_1, y_2, y_3, \dots, y_i, \dots, y_n\} \quad (7)$$

Where $y_i = P^T(x_i - x_{mean})$ and k controls the principal component matrix P and hence controls the dimension of output Y .

On the basis of PCA, we reduced the data of the input layer and the last fully-connected layer to 30 dimensionalities. After PCA, the t-distributed Stochastic Neighbor Embedding (t-SNE) is used for the visualization of the dataset. The t-SNE can evaluate the original data that is entered into the algorithm and find the best way of representing the data using fewer dimensions by matching both distributions. The t-SNE result of the last fully-connected layer is shown in Figure 3(f), which acquires much clearer boundaries of different classes indicating the desirable performance of proposed CNN methods. The discussion of PCA is added in Page 8 and Page 17-18 of the manuscript while the description of t-SNE for visualization is included in Page 9 for your evaluation.

Figure 3. Word and sentence recognition based on the non-segmentation method. (a)-(c) The optimization of CNN structure parameters based on accuracy performance, including Kernel size (a), the number of filters (b), and the number of convolutional layers (c). (d) The final structure of CNN after optimization. The cluster results of word signal from CNN input layer (e) and output layer (f). (g) The confusion map of recognizing 50 words. The cluster results of sentence signal from CNN input layer (h) and output layer (i). (j) The confusion map of recognizing 17 sentences.

5. Compared with Nat. Electron., 3(9), 571-578 (2020) which is mentioned as Ref. 59 in your manuscript, may the authors emphasize the advantages of your work in comparison?

We thank the reviewer for the constructive comment. For Nat. Electron., 3(9), 571-578 (2020), Z. Zhou *et al.* reported a sign-to-speech translation system consisted of stretchable triboelectric fiber-based finger sensors, a wireless printed circuit board

(PCB), and a customized mobile application that is embedded with a machine-learning algorithm. The system achieved a highly accurate recognition of 11 signs, including four numbers, six letters, and one word. Finally, the recognition result was unidirectionally displayed on the mobile phone interface, which paved the way for communication between signers and non-signers. Comparably, our work demonstrated a sign language recognition and communication system assisted by cutting-edge artificial intelligence (AI) and virtual reality (VR) techniques. It successfully achieved the high-accuracy classification of 50 words and 20 sentences. Significantly, the convolutional neural network (CNN) classifier enabled the recognition of never-seen new sentences via new-sequence recombination of word units that split from ever-seen sentences. Besides, the building VR block allowed two-communication of signers and non-signers. To involve in the communication, the signer could perform their familiar sign language and the non-signer directly typed.

Thus, there are two major advances contributed by our work. **First**, although Z. Zhou *et al.* demonstrated incorporating AI to realize the classification of a dozen sign gestures in a highly accurate manner, it is still limited to the identification of simple and discrete signs of numbers/letters/words instead of sentences in sign language. The lack of an effective approach to perceive sentences of sign language hinders the communication of the signer with non-signer, weakening the feeling of social assimilation of the disabled. Accordingly, in addition to words (i.e., single gestures), our CNN could expeditiously differentiate sentences that contain multiple gestures, especially new sentences. The segmentation method split ever-seen/trained sentence signals into basic fragments (i.e., signals of word units) first. The CNN classifier would next recognize all the word elements from the fragment pool. In such a way, the CNN achieved the recognition of never-seen new sentences that were in new-order recombination of words. This segmentation aided AI offers a universal strategy to identify new sentences and to expand the recognizable sign language database.

Second, Ref. 59 could only realize one-way delivery of sign language recognition results, not allowing interaction between the signer and non-signer, which might diminish the recognition system's practicality. In our work, the back-end VR that improved the user experience was developed to act as the bridge of user terminals, transmitting messages back and forth. It was incorporated with the smart glove and AI part to configure a closed-loop recognition and communication system. The speech/hearing impaired could use sign language to interact with the non-signer (i.e., human-to-human interaction). The AI would recognize the generated gesture signals from smart gloves and send them to the client of VR space. At the same time, the non-signer responded through direct typing (i.e., one of the forms of speech language) in the server of VR space. In summary, the capabilities of new sentence recognition and bi-directional communication provide an opportunity to enhance the universality and practicality of the sign language translation system.

6. In section ‘Recognizing new/never-seen sentences upon segmentation approach’, line 329 shows the hierarchy classifier achieves an average correct rate of 86.67% for recognizing new sentences. Although the authors provide the detailed Table S2, it does not directly indicate the average correct rate 86.67%. The authors may add one graph like Figure 4(f) and (i) to clearly show the recognition accuracy of new sentences.

Thank the reviewer for this suggestion. As indicated below, we have added graphs for a more intuitive exhibition of the average correct rate by visualizing the predicted or recognized data shown in Table S2. This graph has been included in Page 9 of Supplementary Materials as Figure S6.

Figure S6. The recognition result of three new sentences where each has been tested five times with five samples. (a) Using single classifier. (b) Using hierarchy classifier. The false prediction area is greatly reduced by using hierarchy classifier.

7. How would the authors evaluate the two approaches mentioned in this work, i.e., the segmentation and non-segmentation methods?

We thank the reviewer for the constructive comment. To illustrate the pros and cons of non-segmentation and segmentation methods, the detailed implementation of these two approaches should be discussed first. **For the non-segmentation method**, each word or sentence is labeled as the independent individual. Then all the words and sentences will be separately trained in the neural network. Upon the completion of training, the CNN will recognize words and sentences independently. With such a regime, either words or sentences essentially are different classes with respect to CNN’s cognition, in which there is no built-up relationship between word units and sentences. **For the strategy of segmentation**, the data sliding window divides the entire sentence signal (800 data points) into fragments, including intact word signals, incomplete word signals, and background signals. The label of fragment split from sentence signal is determined by the principal component. In other words, either word signal or background noise accounts for more than 50% of the sliding window size, and the label will be the number of corresponding words or 'empty' 19 as shown in Figure 4(a). Due to the specific size (200 data points) and sliding step (50 data points) of the sliding window, the entire sentence signal is split into 13 fragments where each one is labeled with a number. Hence, the label of the sentence will be a series of 13 numbers, as illustrated at the top of Figure 4(b). Next, the sentence signals with the label of number sequences are included in the dataset for training. The CNN classifier will go through all the fragments as well as the fragment sequence in sentences. Ultimately, both fragments and reversely reconstructed

sentences by virtue of fragments can be correctly recognized. In particular, the CNN classifier is even endowed with the capability to recognize never-seen sentences that comprise new-order word fragments. The never-seen sentences are not included in the dataset for the training process and hence never learned by the neural network before.

Overall, as the radar map of comparison in Figure S7 shows, **the non-segmentation approach** possesses better performance in recognition accuracy either for words or sentences. However, two following limitations of this means may compromise the universality and practicality of the whole system. **Above all**, owing to the independence of words and sentences, the CNN classifier cannot identify the new sentence, although words in the sentence are seen before and only combined in a new order. **In addition**, when expanding sentence database, the labor-intensive data collection of new sentences and successive training are unavoidable. This kind of independence also leads to increased effort on data collection of words since the CNN model cannot extract and recognize the word signals in the sentence. Regarding **the segmentation method**, in addition to identifying existing sentences in the dataset, the CNN classifier enables the recognition of new sentences. These new sentences comprise new-order word series that is different from the order of existed sentences in the dataset. **Nevertheless**, the segmentation introduces a large amount of random and irregular ‘empty’ signals. It sacrifices the recognition accuracy for both words and sentences. Further research efforts could be committed to optimizing the algorithm framework and improve the recognition accuracy. The above discussion and the below graph are added in Supplementary Materials as Note S2 (Page 10) and Figure S7 (Page 11) for your evaluation.

Figure 4. Word and sentence recognition based on segmentation method which brings the feasibility of new/never-seen sentence recognition. (a) The label table of 19 words (they are from the above-mentioned 50 words) that present in 20 sentences. (b) The schematic diagram of sentence signal segmentation, 'The dog scared me' is taken as an example. (c) The summary table of sentences with category remarks, comprised words, segmented label series, and unique labeled number order. (d) The schematic diagram of single classifier. (e) The confusion map of split element recognition based on sentence signals (recognition accuracy 81.9%). (f) With successful recognition of each element in sentences, the sentence can be inversely reconstructed and recognized at an average correct rate of 79.41% within single classifier. (g) The schematic diagram of the hierarchy classifier. (h) The confusion map of segmented element recognition based on sentence signals (recognition accuracy of 82.8%). (i) With successful recognition of each element in sentences, the sentence can be inversely reconstructed and recognition at an average correct rate of 85.58% within hierarchy classifier. (j) The recognition of 3 new sentences that the CNN model did not learn before, taking 'I lost my dog' as an example. The detailed recognition results can be found in Table S2.

Figure S7. The radar comparison map of two methods (i.e., non-segmentation and segmentation) based on their pros and cons.

- For the VR demonstration in Figure 8, why the signer cannot directly type in the message to communicate with the healthy subject? Since there is a laptop or computer, what would be the necessity of the machine-learning-enabled glove? How is it beneficial for the signers in practical applications?

Thanks a lot for the valuable comment. Indeed, due to born deaf and mute, congenital hearing/speech impaired people, they have many more difficulties acquiring written language than the healthy group. **On the one hand**, they do not have basic cognition about the text that is created by humans and in weak connections with the dynamic world. Even with long-time demanding training, they may not be able to understand this kind of semiotic language very well. ^[6] **On the other hand**, the surrounding is always changing and developing. The auditory sense is considered as the most effective way among the five senses to understanding the dynamic ambient since the sound is intrinsically embedded in the time axis. The hearing/speech disordered group loses the most important feeling that builds contact with surroundings and has to turn to the vision which could also well describe the dynamics. The sign language that comprises different gestures located on the time sequence comes into being and acts as the linkage bridge between the disabled and the real world. ^[7] **Therefore**, as an independent language system, sign language does not evolve and simplify from the text language. The original intention of sign language invention is to benefit the congenital hear/speech impaired population. Furthermore, the disabled could invest less time and effort to learn the gesture language that is much easier for them than written language.

In summary, for the hearing/speech disordered people especially congenitally, sign language is their mother tongue. They are skilled with and preferred their primary language, in which the understanding and expressing of semantics are smoother and more fluent. In comparison, for the signers, the use of written language (i.e., static

symbols for concept expression) is obscure and less convenient due to the loss of hearing and speech which is significant for the auxiliary understanding of the text. Hence, the intelligent recognition system for sign language (wearable sensors, vision-based cameras, etc.) still has great significance to lowering the communication barrier between the signers and non-signers.

9. As the authors mention, there are few publications showing gesture sensing and even sign language recognition based on resistive, capacitive, and triboelectric effects. Recognition of sentences even new sentences of sign language using triboelectric glove sensors with AI integration could bring great significance to the research field. The authors could provide a benchmarking table in Supplementary Materials to summarize the features of various reported works in comparison to current work in terms of function, applications etc. It helps show the uniqueness of current work in an intuitive way.

We thank the reviewer's valuable comment and suggestion. The relevant works for sign language recognition are summarized in the following table. **Above all**, reported sign language recognition based on the glove solution identifies a simple and limited variety of numbers and letters with manual simple feature extraction (e.g., signal amplitude and polarity). Although few works demonstrated the incorporation of AI data analytics to realize the classification of a dozen sign language gestures in high accuracy, it still lacks an effective and practical approach for sentence recognition of sign language which is more significant for the practical communication of signers and non-signers. By contrast, our AI-enabled sign language recognition system achieves the classification of 50 words and 20 sentences. In particular, through splitting sentence signals into word units and recombining these units in new sequences, the proposed CNN architecture renders never-seen/new sentence recognition. It provides a universal methodology to readily expand the sentence database, significantly improving the sign language recognition system's practicality. **Second**, the interfaces (e.g., mobile phone or PC) that sign language recognition results are projected or displayed usually do not realize the two-way communication between signers with non-signers. In other words, they just unidirectionally deliver the translated text or audio to the users. This kind of human-to-human interaction hindrance could greatly limit the practical application of the sign language recognition system. Comparably, by depending on the mutual connected client and server of *Unity*, the signers are allowed to directly use sign language in the client end to participate in the communication process while the healthy person just types in the server end to involve in the interaction. **Last but not least**, typical wearable sensors that rely on resistive and capacitive mechanisms for human status tracking generally need an external power supply to generate signal excitation, inhibiting their further widespread deployment. Owing to simple fabrication, wide material choice, and expeditious dynamic response, the triboelectric nanogenerator (TENG) based wearable sensors, which are recognized as power-compatible and self-sustainable alternatives, are increasingly employed in the human motion recognition. The added table is included in the Supplementary Materials Page 16 as Table S3 for your evaluation.

Table S3. The benchmarking table for comparing with other similar works

Mechanism	Device	Method	Sensor/hand	Gesture	Sentence Locations	Recognizing new sentences	Self-powered	Mutual interaction	Ref
Piezoresistive	Armband	SVM	3	8 letters	-	-	x	x	1
Piezoresistive	Radar+wristband	HSVM	5	4 letters	0	x	x	x	2
Resistive	Finger sensing clin	SVM	5	10 letters	0	x	x	x	3
Resistive	Glove	BNN	9	10 letters	0	x	x	x	4
Resistive	Elastomer Glove	Amplitude	5	5	-	-	x	x	5
Resistive	Glove	Amplitude	5	9 words	0	x	x	x	6
Resistive	Glove	Amplitude	9	26 letters	0	x	x	x	7
Capacitive	Glove	Amplitude	5	5 numbers	-	-	x	x	8
Capacitive	Glove	Amplitude	5	36 letters	0	x	x	x	9
Capacitive	Wristband	Amplitude	15	15 words	0	x	x	x	10
Capacitive	Glove	XRF	16	10 numbers	0	x	x	x	11
Capacitive	Finger sensing clin	Amplitude	4	6 numbers	-	-	x	x	12
Ionic	Glove	Amplitude	8	6 words	0	x	V	x	13
Piezoelectric	Glove	Amplitude	5	6 numbers	0	x	V	x	14
Triboelectric	Glove	Amplitude	5	6 numbers	-	-	V	x	15
Triboelectric	Glove	Amplitude	6	7 numbers	-	-	V	x	16
Triboelectric	Glove	Amplitude	5	5 letters	0	x	V	x	17
Triboelectric	Glove	SVM	5	11 letters	0	x	V	x	18
Triboelectric	Glove	CNN	7	50 words	20	V	V	V	*

Note: (H)SVM: (hierarchical) support vector machine; BNN: binary neural network; XRF: extremely randomized trees; CNN: convolutional neural network. (*This work)

Reviewer #3 (Remarks to the Author):

In this manuscript, the authors propose a sign language recognition and communication system, including gloves integrated with triboelectric sensors, AI block and VR interactive interfaces. AI analysis of TENG sensor data is very important for TENG and AI application communities. I recommend the manuscript to publish after major revision. There are several comments as the following:

Thanks a lot for the reviewer's comments. The following is our point-to-point response.

1. There are two main sign language recognition approaches: image-based and sensor-based. [IEEE transactions on human-machine systems, 2014, 44(4): 551-557] For image-based systems, signers use simple devices. For sensor-based systems, smart gloves need personalized design, which are expensive. TENG can overcome the challenge of cost. Compared with image-based technology, authors should provide experiments and discuss the advantages of TENG sensor-based systems.

We appreciate the reviewer for the valuable comment and suggestion. To clarify the advantages of sensor-based system, we have done the additional test about the accuracy performance of visual images for gesture recognition. The recognition results of six representative gestures are shown in Figure S8 under varying light conditions (493, 275, and 13 lux). For each light condition, 50 trials of each gesture (300 trials in total) are carried out for image-based recognition. Figure S8 (b)-(i-iii) indicate a dramatically decayed recognition accuracy from 98.33% to 58.33% when the room light fades. Visual image/video recognition efficiency has been well-known limited by environmental interferences such as occlusions and especially light conditions.^[5] In addition, sign language involves the motions of upper limbs as well as human faces. When the image-based system captures gesture information, the exposure of facial information to camera may arise the issue of privacy disclosure.

For sensor-based human gesture recognition system, wearable sensors are typically less bulky, flexible and provide intimate contact with the user for high-quality data acquisition and high-accurate recognition that is comparable with its image system counterpart. The sensor-based system is considered as one of the approaches to overcome the drawbacks of image recognition. On the one hand, the sensor-based systems are not affected by varying luminance and can work well even under an entirely dark condition with higher environmental tolerance. On the other hand, they can mitigate the privacy issue in a cost-effective way owing to no need for individual information collection such as facial characteristics. The above relevant discussion and below figure are included in Page 12 and Page 13 of Supplementary Materials as Note S3 and Figure S8 for your evaluation.

Figure S8. (a) The gesture image of ‘Love’ under three light conditions. (b) (i-iii) The accuracy of the image recognition under different light conditions (493, 275, and 13 lux). (c) Accuracy degradation with decreased brightness.

- Traditional commercial sensors are good candidate to obtain high-precision data. The author should compare TENG sensors with traditional commercial sensors? How to redesign AI and communication system to overcome the limitation of TENG, for example, accuracy and the principle of orthogonality of the sensor signal? The authors should add experiments to provide the important properties of sensors.

Thanks a lot for the reviewer’s valuable comment and suggestion. **For comparison with commercial sensors**, the inertial measurement unit (IMU) is the most common commercial product for gesture recognition besides the visual image/video based system. [8] The intensive effort has been devoted to the development of IMU-based gesture recognition system. For example, Mummadi, C. *et al.* reported a glove prototype with embedded 9-axis IMU sensors for sign language recognition. As shown in Figure 3 of their work, by relying on the attitude angle calculated from fused data from accelerometers and gyroscopes, the IMU-integrated glove can identify five simple gestures such as hello and goodbye. [9] Notably, it leads to a

limited variety of recognized gestures (i.e., several simple gestures) due to only depending on the single feature or index (e.g., attitude angle). Of course, the leveraging of artificial intelligence (AI) with IMU sensors is also explored for multiple feature extraction and gesture classification. Khomami, S. *et al.* proposed a wearable device based on surface electromyography (sEMG) and IMU sensor. Twenty-five highest-ranked features of two modalities (sEMG, IMU) were extracted and classified by the K-nearest neighbor (KNN) classifier among 22 sign language gestures. ^[10] Although increased features improve the number of discriminated gestures, it remains in recognition of discrete and simple letters/number/words of sign language and needs furthered improvement toward practical sentence interpretation. **In our case, the entire original data of TENG sensors is thrown into the AI processing pool rather than limited feature extraction, which helps redesign the AI system with high precision.** The convolutional neural network (CNN) becomes more and more intelligent through numerous updating and correction in the learning process, in which substantial and comprehensive features are captured by the CNN architecture.

Regarding the limitations of the TENG sensor, as we know, **humidity** is considered as **one of the major factors to diminish the output performance of triboelectric sensors**, making the triboelectric signal random and incomplete. Thus, the additional experiment has been carried out to verify the artificial intelligence (AI) enabled sensor-based recognition system immune to the typical environment interference for TENG (i.e., humidity). In detail, 600 trials for six representative gestures (i.e., love, you, now, want, work, yourself with 100 trials for each) are repeatedly performed for five days which are in different relative humidity (RH). For the training dataset, it is distinctly set as the data samples of Day 1 (RH:40%), Day 1 (RH:40%) + Day 2 (RH:50%), Day 1 (RH:40%) + Day 2 (RH:50%) + Day 3 (RH:65%), and Day 1 (RH:40%) + Day 2 (RH:50%) + Day 3 (RH:65%) + Day 4 (RH:75%). As shown in Figure R2, the test accuracy of Day 5 (RH: 57%) dramatically increases from 42.67% to 90%, in which the data samples of Day 5 are recognized by the CNN architecture that is trained by the above listed datasets. This result proves the TENG sensor based system integrated with AI data analytics can overcome the issue of environment variation induced signal instability as long as the training set contains data samples of several different surroundings and becomes more generalized.

This solution has great feasibility for implementation owing to the following two reasons. First, there is no need for labor-intensive data collection which is from too many ambient conditions. The necessary requirement lies in the reasonable selection of environmental conditions which the training set is collected from. It is acceptable that the chosen surrounding conditions for training set acquisition cover the usual working environment of sensors. In our case, two indoor moist limits RH40% and RH75% in Singapore, are considered as possible candidates plus another two middle-range RH50% and RH65% since the general indoor humidity for sensor operation locates around RH60%-RH65% with a stable ventilation system. Depending on data samples from these four statuses, the trained CNN achieves a high accuracy of 90% when recognizing samples from Day 5 (RH57%). In addition, the recognition results in Figure 3 and Figure 4 of the main text are also based on the training samples that come from different days with varied surroundings.

Because the intact data of sensor signals is included in the training set, CNN can actually seek the common features from unwanted signal interference which derives varied environments via continuous learning process, and further consider signal variations as the inevitable systematic error. In such a way, the behavior and thinking of AI are more and more similar to the human brain. It can catch essential features (e.g., capturing critical signal shapes behaving like the naked eyes of humans) and pay less attention to unnecessary fluctuation such as signal amplitude variation induced by humidity.

Likewise, the orthogonality of the sensor signal, which is also called the crosstalk of sensor channels, is **considered as another limitation of the TENG sensor**. It is widely observed for triboelectric sensors, in which our sensors are also not the exception. As shown in Figure R3, the crosstalk between channels is comparably negligible with the signal. Due to the crosstalk universally existing in sensor signals of every gesture, the CNN can also find the shared points from such crosstalk signals and further categorize them into the system error. In general, the recognition based on the original sensor data (e.g., 1D CNN) is superior to the dependence on the purposely extracted features in terms of solving the system fluctuation problem. Besides, from the aspect of the hardware component, improved system design (e.g., circuit, device packing) could reduce the crosstalk as well.

Lastly, the limitation that only recognizes simple gestures needs to be solved. Most works generally depend on amplitude or polarity to realize the recognition of a limited number of gestures. It is far from the sentence requirement of practical communication between the hearing/speech impaired and healthy persons. Thus, we propose **the segmentation method to achieve sentence especially new sentence recognition**. The data sliding window divides the entire sentence signal (800 data points) into 13 fragments, including intact word signals, incomplete word signals, and background signals. The label of fragments that split from sentence signal is determined by the major component. In other words, either word signal or background noise accounts for more than 50% of the sliding window size, and the label will be the number of corresponding words or 'empty' 19 as shown in Figure 4(a). Hence, the label of the sentence will be a series of 13 numbers, as illustrated at the top of Figure 4(b). The CNN classifier will go through all the fragments as well as the fragment sequence in sentences. Ultimately, both fragments and reversely reconstructed sentences can be correctly recognized. In particular, the CNN classifier is endowed with the capability to recognize never-seen sentences that comprise new-order word fragments (Figure 4(j)). The never-seen sentences are not included in the dataset for the training process and hence never learned by the neural network before. It provides a universal methodology to readily expand the sentence database, greatly improving the sign language recognition system's practicality.

Figure 3. Word and sentence recognition based on the non-segmentation method. (a)-(c) The optimization of CNN structure parameters based on accuracy performance, including Kernel size (a), the number of filters (b), and the number of convolutional layers (c). (d) The final structure of CNN after optimization. The cluster results of word signal from CNN input layer (e) and output layer (f). (g) The confusion map of recognizing 50 words. The cluster results of sentence signal from CNN input layer (h) and output layer (i). (j) The confusion map of recognizing 17 sentences.

Figure 4. Word and sentence recognition based on segmentation method which brings the feasibility of new/never-seen sentence recognition. (a) The label table of 19 words (they are from the above-mentioned 50 words) that present in 20 sentences. (b) The schematic diagram of sentence signal segmentation, 'The dog scared me' is taken as an example. (c) The summary table of sentences with category remarks, comprised words, segmented label series, and unique labeled number order. (d) The schematic diagram of single classifier. (e) The confusion map of split element recognition based on sentence signals (recognition accuracy 81.9%). (f) With successful recognition of each element in sentences, the sentence can be inversely reconstructed and recognized at an average correct rate of 79.41% within single classifier. (g) The schematic diagram of the hierarchy classifier. (h) The confusion map of segmented element recognition based on sentence signals (recognition accuracy of 82.8%). (i) With successful recognition of each element in sentences, the sentence can be inversely reconstructed and recognition at an average correct rate of 85.58% within hierarchy classifier. (j) The recognition of 3 new sentences that the CNN model did not learn before, taking 'I lost my dog' as an example. The detailed recognition results can be found in Table S2.

Figure R2. The Day-5 accuracy of sensor-based gesture recognition with the training set of (a) Day 1 (RH:40%), (b) Day 1 (RH:40%) + Day 2 (RH:50%), (c) Day 1 (RH:40%) + Day 2 (RH:50%) + Day 3 (RH:65%), (d) Day 1 (RH:40%) + Day 2 (RH:50%) + Day 3 (RH:65%) + Day 4 (RH:75%).

Figure R3. (a) The real-time signal when the pressure is only applied on the sensor of the middle finger of the right hand. The purple rectangle indicates the crosstalk, (b) the crosstalk map of each channel with the rest 14 channels.

- AI technology needs to recover information and improve the accuracy of information by learning the characteristics of complete information. [Nano Energy, 2021: 105887] Especially, the data obtained by the TENG sensor is random and low accuracy. The authors should provide the detailed discussion of TENG sensor-based systems.

Thanks a lot for the reviewer's valuable comment and suggestion. Generally, there are three major factors that lead to the instability of triboelectric signals. **First**, due

to the randomness and fluctuation of biomechanical motions, the AC electrical outputs of TENG are usually irregular and not stable. ^[11] The precision of various sensing or recognition applications is hence compromised when simply depending on amplitude/bandwidth/polarity. As the developed strategy to mitigate this issue, the design electrode pattern could generate stable numbers of peaks with excellent reliability and robustness since recognition of signal patterns is regardless of absolute amplitude and thereby not affected by sliding speed/force. ^[12] **Second**, for the electromagnetic coupling and disturbance of surrounding objects, grounding packaging is widely considered as a common strategy to mitigate this concern. Besides, the crosstalk could be decreased by the carefully designed and arranged electrode wires. **Third**, environmental conditions such as humidity cannot be bypassed when considering the triboelectric signal instability. Encapsulation, material modification for better tolerance, and the extra reference sensor are frequently adopted to relieve the environmental restriction. ^[13-15] **Finally**, In the era of big data, benefitting from the strong computation and learning capabilities, AI offers a new opportunity to obtain a highly accurate analysis and decision. As the response of the last comment mentions, a generalized training set with increased data sample from more conditions could be a universal guideline to ease the negative effect of these external interferences on the precise identifying of sensing signals. In addition to the training set generalization, the innovation of AI algorithm could be another way to improve accuracy performance even with environmental condition variation.

References

- [1] Zhou, Z. *et al.* Sign-to-speech translation using machine-learning-assisted stretchable sensor arrays. *Nat. Electron.*, **3**(9), 571-578 (2020).
- [2] Araromi, O. A. *et al.* Ultra-sensitive and resilient compliant strain gauges for soft machines. *Nature*, **587**(7833), 219-224 (2020).
- [3] Suzuki, K. *et al.* Rapid-response, widely stretchable sensor of aligned MWCNT/elastomer composites for human motion detection. *ACS Sens.*, **1**(6), 817-825 (2016).
- [4] Liu, S., Zhang, J., Zhang, Y., & Zhu, R. A wearable motion capture device able to detect dynamic motion of human limbs. *Nat. Commun.*, **11**(1), 1-12 (2020)
- [5] Wang, M. *et al.* Gesture recognition using a bioinspired learning architecture that integrates visual data with somatosensory data from stretchable sensors. *Nat. Electron.*, **3**(9), 563-570 (2020).
- [6] Emmorey, K. Language, cognition, and the brain: Insights from sign language research. *Psychology Press* (2001).
- [7] Bellugi, U., & Fischer, S. A comparison of sign language and spoken language. *Cognition*, **1**(2-3), 173-200 (1972).
- [8] Esposito, D. *et al.* A piezoresistive array armband with reduced number of sensors for hand gesture recognition. *Front. in neurorobot.*, **13**, 114 (2020).
- [9] Lei, L., & Dashun, Q. Design of data-glove and Chinese sign language recognition system based on ARM9. *12th IEEE ICEMI*, **3**, 1130-1134 (2015).
- [10] Khomami, S. A., & Shamekhi, S. Persian sign language recognition using IMU and surface EMG sensors. *Measurement*, **168**, 108471 (2021).
- [11] Wu, C., Wang, A. C., Ding, W., Guo, H., & Wang, Z. L. Triboelectric nanogenerator: a foundation of the energy for the new era. *Adv. Energy Mater.*, **9**(1), 1802906 (2019).

- [12] Shi, Q., & Lee, C. Self-powered bio-inspired spider-net-coding interface using single-electrode triboelectric nanogenerator. *Adv. Sci.*, **6**(15), 1900617 (2019).
- [13] Mule, A. R., Dudem, B., Graham, S. A., & Yu, J. S. Humidity sustained wearable pouch-type triboelectric nanogenerator for harvesting mechanical energy from human activities. *Adv. Funct. Mater.*, **29**(17), 1807779 (2019).
- [14] Wen, F. *et al.* Machine learning glove using self-powered conductive superhydrophobic triboelectric textile for gesture recognition in VR/AR applications. *Adv. Sci.*, **7**(14), 2000261 (2020).
- [15] Zhu, M. *et al.* Self-powered and self-functional cotton sock using piezoelectric and triboelectric hybrid mechanism for healthcare and sports monitoring. *ACS Nano*, **13**(2), 1940-1952 (2019).

REVIEWERS' COMMENTS

Reviewer #1 (Remarks to the Author):

The authors have revised the manuscript carefully according to the reviewer's comments. The quality of this work is greatly improved. The reviewer is satisfied with the revision and has no more other suggestion for this work. I think that this manuscript should be accepted by the journal.

Reviewer #2 (Remarks to the Author):

The authors have addressed all my concerns and the MS is in a good shape and it can be accepted for publication without change.

Reviewer #3 (Remarks to the Author):

I recommend the revised manuscript to publish.

REVIEWERS' COMMENTS

Reviewer #1 (Remarks to the Author):

The authors have revised the manuscript carefully according to the reviewer's comments. The quality of this work is greatly improved. The reviewer is satisfied with the revision and has no more other suggestion for this work. I think that this manuscript should be accepted by the journal.

We thank a lot for reviewer's recommendation and the great efforts in the previous review.

Reviewer #2 (Remarks to the Author):

The authors have addressed all my concerns and the MS is in a good shape and it can be accepted for publication without change.

We thank a lot for reviewer's recommendation and the great efforts in the previous review.

Reviewer #3 (Remarks to the Author):

I recommend the revised manuscript to publish.

We thank a lot for reviewer's recommendation and the great efforts in the previous review.